# Loss of mTORC1 signalling impairs β-cell homeostasis and insulin processing

Manuel Blandino-Rosano[1], Rebecca Barbaresso[2], Margarita Jimenez-Palomares[2], Nadejda Bozadjieva[2], Joao Pedro Werneck-de-Castro[1], Masayuki Hatanaka[3], Raghavendra G. Mirmira[3], Nahum Sonenberg[4], Ming Liu[2], Markus A. Rüegg[5], Michael N. Hall[5] & Ernesto Bernal-Mizrachi[1,6]

Deregulation of mTOR complex 1 (mTORC1) signalling increases the risk for metabolic diseases, including type 2 diabetes. Here we show that β-cell-specific loss of mTORC1 causes diabetes and β-cell failure due to defects in proliferation, autophagy, apoptosis and insulin secretion by using mice with conditional (*βraKO*) and inducible (*MIP-βraKO^{f/f}*) *raptor* deletion. Through genetic reconstitution of mTORC1 downstream targets, we identify mTORC1/S6K pathway as the mechanism by which mTORC1 regulates β-cell apoptosis, size and autophagy, whereas mTORC1/4E-BP2-eIF4E pathway regulates β-cell proliferation. Restoration of both pathways partially recovers β-cell mass and hyperglycaemia. This study also demonstrates a central role of mTORC1 in controlling insulin processing by regulating cap-dependent translation of carboxypeptidase E in a 4EBP2/eIF4E-dependent manner. Rapamycin treatment decreases CPE expression and insulin secretion in mice and human islets. We suggest an important role of mTORC1 in β-cells and identify downstream pathways driving β-cell mass, function and insulin processing.

[1] Division of Endocrinology, Diabetes and Metabolism, University of Miami, Miller School of Medicine, Miami, Florida 33136, USA. [2] Department of Internal Medicine, Division of Metabolism, Endocrinology and Diabetes, Brehm Center for Diabetes Research, University of Michigan Medical Center, Ann Arbor, Michigan 48105, USA. [3] Department of Pediatrics, Indiana University School of Medicine, Indianapolis, Indiana 46202, USA. [4] Department of Biochemistry, McGill University, Montreal, Quebec H3A 1A3, Canada. [5] Biozentrum, University of Basel, CH-4056 Basel, Switzerland. [6] Miami VA Health Care System, Miami, Florida 33136, USA. Correspondence and requests for materials should be addressed to E.B.-M. (email: EbernalM@med.miami.edu).

mTOR pathway links upstream nutrient availability and growth factor signalling to control metabolism, cell growth, proliferation and protein synthesis by phosphorylation of key components[1–3]. Growing evidence indicates that mTOR signalling pathway is deregulated in human diseases, including type 2 diabetes[4–6]. The importance of mTOR signalling in regulation of insulin sensitivity has been demonstrated[5]. However, how alterations of this pathway in β-cells contribute to the pathogenesis of diabetes is less understood.

mTOR functions in two distinct multi-protein complexes termed mTOR complex 1 (mTORC1) and mTORC2. mTORC1 constitutes the rapamycin-sensitive arm of mTOR signalling and contains six components, including mTOR, mLst/GβL, Deptor, Tti1/Tel2 complex, Raptor and PRAS40 (ref. 7). Raptor and PRAS40 are specific to the mTORC1 complex, and deletion of Raptor inactivates this complex. mTORC1 controls cell size, proliferation, ribosomal biogenesis, protein translation and autophagy by modulating eIF4E-binding proteins (4E-BP1, 2 and 3) and ribosomal protein S6 kinases (S6K1 and 2) and ULK among others[1,3]. S6K phosphorylates downstream substrates, such as ribosomal S6 protein and eIF4B, to promote mRNA translation and synthesis of ribosomes. Phosphorylation of 4E-BPs triggers their release from eIF4E and initiates cap-dependent translation of mRNAs with complex 5′-untranslated region (UTR) structures. Loss of mTORC1 signalling in liver, muscle and adipocytes by tissue-specific deletion of raptor demonstrate that mTORC1 contributes to the control of metabolism and energy homeostasis in a tissue-specific manner[8–11]. In addition, the use of mTORC1 inhibitors (rapamycin) and analogues (rapalogs) has provided information about the role of this pathway in human disease and further suggest that this pathway is involved in human diabetes[12].

The current studies uncover the role of endogenous mTORC1 signalling in β-cells using mice with conditional raptor deletion in β-cells (βraKO). This deletion results in β-cell failure and diabetes by reduction in proliferation, cell size and survival. Using mice with inducible deletion of raptor in mature β-cells, we identify a novel role of mTORC1 on insulin secretion. To investigate mechanistically how mTORC1 inactivation induces β-cell failure, we perform genetic reconstitution of 4E-BP1-2/eIF4E or S6K activity in βraKO mice. Genetic reconstitution of 4E-BPs/eIF4E and S6K signalling in βraKO mice shows that mTORC1 orchestrates a signalling response to regulate cell survival, β-cell mass and insulin secretion. Moreover, we find a novel role for the mTORC1/4E-BP2/eIF4E arm in the regulation of insulin processing by controlling cap-dependent translation of carboxypeptidase E (CPE). Finally, rapamycin treatment in mice and human islets recapitulates the effect of mTORC1 on CPE, suggesting that this mechanism could be relevant to humans treated with this agent.

## Results

**Disruption of mTORC1 in β-cells causes diabetes.** To inactivate mTORC1 function, we generated mice with homozygous deletion of raptor in β-cells by crossing raptor[f/f] with Rip-Cre mice (βraKO)[8,13]. βraKO islets exhibited a reduction in ~80% of Raptor levels leading to a decrease in the phosphorylation of the mTORC1 targets 4E-BP1 and S6 protein (Fig. 1a and Supplementary Fig. 1). The remaining Raptor and p-S6 detected in isolated islets are likely due to immunoreactivity coming from non-β-cells, islet culture conditions with growth factors and phosphorylation of S6K by other pathways[14–17]. The fall in p-S6 in β-cells was also observed in pancreas sections (Supplementary Fig. 2a). Furthermore, βraKO mice when crossed to CAG-GFP reporter mice showed that 95% of insulin-positive

cells were also positive for green fluorescent protein (GFP) indicating that this Cre-recombinase line induced recombination in the majority of β-cells (Supplementary Fig. 2b). GFP fluorescence was not observed in glucagon cells or in other areas of the pancreas at 30 days of age suggesting there was no conversion to other cell fates (Supplementary Fig. 2b). These studies clearly demonstrate successful inactivation of mTORC1 signalling specific to β-cells. Assessment of glucose homeostasis showed that random blood fed glucose and insulin levels in βraKO mice were normal during the first 3 weeks of life (Fig. 1b,c). However, glucose levels in male and female mice progressively increased and these mice exhibited severe diabetes in adulthood, accompanied by hypoinsulinaemia (Fig. 1b,c and Supplementary Fig. 3a). In contrast, deletion of only one raptor allele (heterozygous mice) displayed normal fed glucose levels, glucose tolerance and insulin sensitivity (Fig. 1b and Supplementary Fig. 3b,c). These studies suggested that impaired mTORC1 signalling leads to hyperglycaemia and diabetes.

βraKO mice showed progressive reduction in β-cell mass by decreases in proliferation, survival and cell size. β-cell mass is a critical determinant for glucose homeostasis in rodents and humans. βraKO mice were born with normal β-cell mass (Supplementary Fig. 4a–d). However, β-cell mass was reduced by ~40% (Fig. 1d) due to less proliferation and greater apoptosis at 30 days of age (Fig. 1e,f), but this was not sufficient to alter glucose tolerance (Supplementary Fig. 4e) or random glucose levels as described above (Fig. 1b). Glucose-induced β-cell proliferation was also diminished in 30-day-old βraKO islets (Fig. 1g). The reduction in β-cell mass was not associated to changes in pancreatic glucagon content ($0.2812 \pm 0.0308$ versus $0.2806 \pm 0.029$, ng per μg protein, $n = 4$, $P = 0.0571$, data expressed as means ± s.e.m.; nonparametric U-test, Mann–Whitney), although staining for non-β-cells showed that the number of glucagon and somatostatin cells appeared to be increased (Supplementary Fig. 5a–d). This apparent increase in other pancreatic cells has been described in other previous studies of β-cell loss[18]. Importantly, there were no signs of impaired β-cell maturation as demonstrated by similar staining for PDX1, MafA, Pax6 and Glut2 in βraKO mice (Supplementary Fig. 6a–d). Remarkably, β-cell mass is reduced by 80% at 90 days of age, and these changes were accompanied by reduced cell size measured by β-catenin/insulin double staining decreased (Fig. 1h,i and Supplementary Fig. 7). Thus, mTORC1 is necessary for the maintenance of postnatal β-cell mass by controlling apoptosis, β-cell size and proliferation.

**mTORC1 inactivation in mature β-cells impairs secretion.** The results obtained with the βraKO mice indicate that mTORC1 is critical to maintenance of postnatal β-cell mass. To dissociate the developmental and postnatal role of mTORC1, we generated a mouse with tamoxifen (TMX)-inducible deletion of raptor (MIP-βraKO[f/f])[19]. Islets from 2-month-old MIP-βraKO[f/f] mice displayed reduced levels of Raptor, p-S6 and p-4E-BP1 at 2 weeks post TMX injection (Supplementary Fig. 8a). Similar to βraKO mice, MIP-βraKO[f/f] mice exhibited normal weight ($28.12 \pm 0.966$ versus $26.18 \pm 0.726$ g, $n = 9$, $P = 0.1283$) and were hyperglycaemic and hypoinsulinaemic after 4 and 8 weeks post TMX injection (Fig. 1j,k). While glucose tolerance was normal before TMX injection (Fig. 1l), MIP-βraKO[f/f] mice developed impaired glucose tolerance (Supplementary Fig. 8b and Fig. 1m) and defective glucose-stimulated insulin secretion (GSIS) in vivo and in vitro after TMX injection (Fig. 1n,o). Importantly, MIP-βraKO[f/f] mice injected with corn oil had similar glucose tolerance to control mice (Supplementary Fig. 8b and Fig. 1m). Similar to MIP-βraKO[f/f], islets from 4-week-old βraKO mice also exhibited

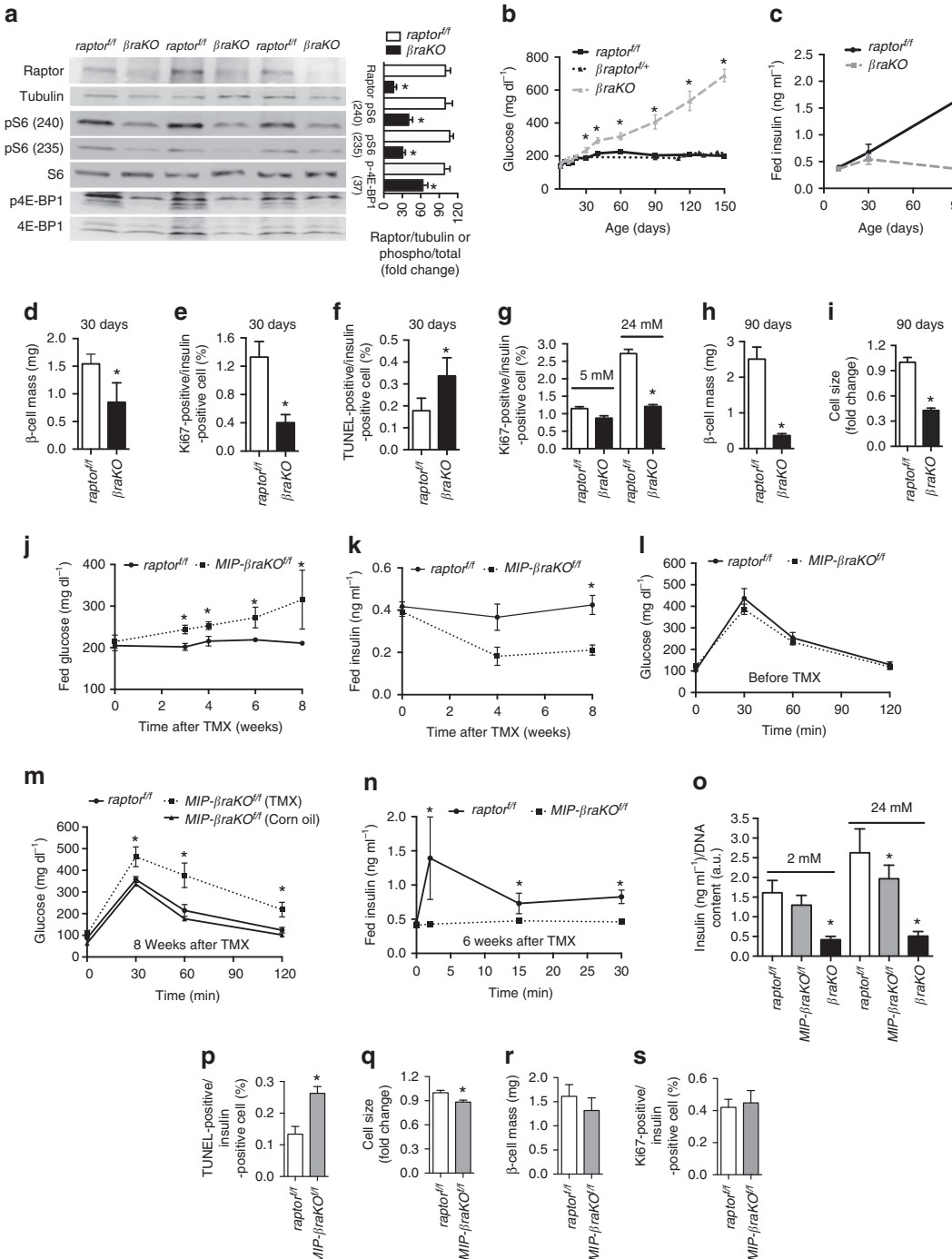

**Figure 1 | Glucose homeostasis and islet morphology in mice with constitutive or inducible loss of mTORC1 function. (a)** Immunoblotting and quantification for Raptor, pS6 (S235 and S240), total S6, p4E-BP1 (T37), total 4E-BP1 and tubulin in islets from *raptor*$^{f/f}$ and *βraKO* mice at 30 days of age. A representative image from four independent experiments is included and each lane shows the expression levels from one mouse. **(b)** Random fed blood glucose levels in *raptor*$^{f/f}$, *βraptor*$^{f/+}$ and *βraKO* mice during the first 150 days ($n = 10$). **(c)** Random fed serum insulin levels in *raptor*$^{f/f}$ and *βraKO* mice at 7, 30 and 90 days of age ($n = 6$). **(d–f)** Assessments of β-cell mass, proliferation by Ki67 and apoptosis by TUNEL in 30-day-old *raptor*$^{f/f}$ and *βraKO* mice ($n = 6$). **(g)** Proliferative responses to glucose in dispersed islets from 30-day-old *raptor*$^{f/f}$ and *βraKO* mice cultured independently for 24 h in 5 or 24 mM glucose. Approximately 1,000 β-cells were counted per experiment ($n = 4$). **(h,i)** Assessment of β-cell mass and cell size analysis in *raptor*$^{f/f}$ and *βraKO* mice at 90 days ($n = 6$). **(j,k)** Random fed blood glucose and insulin levels before and after 4 and 8 weeks post TMX injection (TMX administered at 2 months of age). **(l,m)** Intraperitoneal glucose tolerance test before and after 8 weeks post TMX injection. *MIP-βraKO*$^{f/f}$ mice injected with corn oil are shown in these experiments as controls. **(n)** GSIS (2 versus 24 mM) in islets from *raptor*$^{f/f}$ and *MIP-βraKO*$^{f/f}$ mice at 6 weeks after TMX injection ($n = 5$; **j–n**). **(o)** GSIS *in vitro* using isolated islets from 2-month-old *raptor*$^{f/f}$, *MIP-βraKO*$^{f/f}$ and 1-month-old *βraKO*. *MIP-βraKO*$^{f/f}$ and controls mice were followed for 4 weeks post TMX injection before islet isolation (TMX injection administered at 1 month of age; $n = 4$). **(p–s)** Assessments of apoptosis by TUNEL, cell size, β-cell mass and proliferation by Ki67 staining in *raptor*$^{f/f}$ and *MIP-βraKO*$^{f/f}$ mice at 8 weeks after TMX injection (120 days of age; TMX administered at 2 months of age; $n = 5$). Measure of cell size expressed as a fold change. Data expressed as means ± s.e.m., *$P < 0.05$. Nonparametric *U*-test (Mann–Whitney).

impaired GSIS (Fig. 1o). Assessment of pancreas morphology showed that *MIP-βraKO^{f/f}* mice also exhibited increased apoptosis and decreased cell size (Fig. 1p,q) with no significant alteration in β-cell mass or proliferation at 8 weeks post TMX injection (Fig. 1r,s). Interestingly, apoptosis is already triggered in *MIP-βraKO^{f/f}* mice 2 weeks after TMX as attested by increased cleaved caspase 3 (Supplementary Fig. 8c). No changes in the proapoptotic protein BimEL were noted (Supplementary Fig. 8c). Given the known role of mTORC1 in autophagy and the connection between autophagy and cell death[20,21], we investigated autophagy in the *MIP-βraKO^{f/f}* mice. These studies showed that LC3-II was increased, whereas p-ULK was decreased in *MIP-βraKO^{f/f}* islets at 2 weeks post TMX injection suggesting the presence of autophagy in this model (Supplementary Fig. 8c). These studies demonstrate that *MIP-βraKO^{f/f}* and *βraKO* mice share similar survival, cell size and metabolic phenotypes. Also, the reduction in insulin secretion with virtually normal β-cell mass in the *MIP-βraKO^{f/f}* indicates that mTORC1 also controls insulin secretion *per se*.

**raptor deletion in β-cells induces autophagy**. We next investigated the mechanisms responsible for β-cell loss in *βraKO* mice. Normal levels of C/EBP homologous protein (CHOP) and Bax suggest that endoplasmic reticulum (ER) stress and proapoptotic proteins were not involved (Supplementary Fig. 9a–c). Given the known role of mTORC1 in autophagy, we next decided to investigate the role of autophagy as a mechanism for β-cell failure in *βraKO* mice. We observed a greater number of autophagic vacuoles containing insulin granules, also known as crinophagy[22], in *βraKO* mice compared to controls (Fig. 2a,b). To detect autophagy *in vivo*, we crossed control and *βraKO* mice with mice expressing a molecular marker of autophagy (LC3-GFP)[23]. Accumulation of the LC3-GFP protein is highly specific for autophagosomes and autophagy[24]. Dispersed β-cells from control mice showed low cytoplasmic GFP fluorescence (a read-out of LC3 expression; Fig. 2c). In contrast, cytoplasmic GFP fluorescence was elevated in the majority of *βraKO* β-cells. Quantification of β-cells with LC3-GFP puncta formation confirmed the presence of autophagosome and autophagy in *βraKO* mice at 40 days of age (Fig. 2c).

**Autophagy contributes to the *βraKO* phenotype**. Next, we treated islets of *βraKO* mice with NH₄Cl, an inhibitor of autophagy that prevents endosomal acidification and blocks autophagic flux[25]. Again, *βraKO* mice showed higher basal levels of LC3-II than control mice (Fig. 3a). NH₄Cl treatment resulted in accumulation of p62 and LC3-II protein in both *βraKO* and control mice showing that autophagy was successfully blocked (Fig. 3a). However, the accumulation was greater in *βraKO* mice, confirming the increased autophagy flux in these mice (Fig. 3a). Notably, pharmacological blockage of autophagy by NH₄Cl decreased apoptosis in *βraKO* β-cells (Fig. 3b). Then, we expanded our studies by *in vivo* treatment of *βraKO* mice with two other autophagy inhibitors, chloroquine (CQ) and 3-methyladenine (3MA) starting at 18 days of life. As expected, there was accumulation of p62 in islets from 3MA- and CQ-treated groups providing evidence that autophagy was inhibited by the treatment (Fig. 3c and Supplementary Fig. 10). CQ- and 3MA-treated *βraKO* mice improved glucose levels during the first 60 days of life (Fig. 3d,e). In addition, 3MA treatment doubles the β-cell mass in *βraKO* mice restoring β-cell mass to ∼70–75% of control mice (Fig. 3f), as a result of increased β-cell proliferation and inhibition of apoptosis (Fig. 3g,h). On the other hand, insulin secretion in islets from

*βraKO* mice was not improved after inhibition of autophagy (Fig. 3i).

**Gain of S6K and 4E-BP2 signalling improves glucose levels**. The previous studies showed that mTORC1 is important for the maintenance of postnatal β-cell mass. We next designed experiments to assess contribution of mTORC1 downstream targets to regulation of β-cell mass, survival and function. We selectively restored S6K pathway in *βraKO* mice by crossing with a mouse model with gain of function of S6K (*caS6K*) (*βraKO;caS6K*). Increased p-S6 immunostaining in islets from these mice demonstrated that S6K activity was augmented (Supplementary Fig. 11). Published data show that activation of S6K signalling by overexpressing an active S6K-mutant in β-cells improves β-cell function but not mass[26]. Reconstitution of eIF4E activity was obtained by crossing *βraKO* with mice with global inactivation of eIF4E inhibitors 4E-BP1 or 4E-BP2 (*βraKO;Eif4ebp1^{−/−}* and *βraKO;Eif4ebp2^{−/−}*, respectively). 4E-BP1 and 4E-BP2 are expressed in islets, and deletion of either does not result in a compensatory increase in the level of the other protein[27]. As previously described, examination of glucose homeostasis in *Eif4ebp1^{−/−}* demonstrated improved glucose clearance, and this phenotype resulted from enhanced insulin sensitivity[28]. In contrast to *Eif4ebp1^{−/−}* mice, no alterations in insulin sensitivity were observed in *Eif4ebp2^{−/−}* mice[27]. We recently showed that *Eif4ebp2^{−/−}* and not *Eif4ebp1^{−/−}* mice exhibit improved glucose homeostasis by increase in β-cell mass and proliferation[27]. Assessment of glucose levels in these genetic models showed that hyperglycaemia measured by random fed blood glucose was partially rescued by restoration of 4EBP2/eIF4E or S6k signalling in *βraKO* mice, but not by 4EBP1/eIF4E (Fig. 4a,b). Remarkably, reactivation of 4E-BP2/eIF4E together with S6K signalling was sufficient to maintain normoglycaemia for the first 60 days (Fig. 4b). After that, *βraKO;caS6K;Eif4ebp2^{−/−}* mice exhibited a gradual increase in glucose levels albeit still better than reconstitution of either pathways alone (Fig. 4a,b). Therefore, it is noteworthy that hyperglycaemia in *βraKO* mice was not fully rescued by any of the genetic reconstitution models. The improvement in glucose was not explained by alterations in insulin sensitivity (Fig. 4c). In contrast to fed glucose levels, re-establishing mTORC1 signalling through 4EBP1/eIF4E, 4EBP2/eIF4E and/or S6K normalized glucose levels after 16 h fasting (Fig. 4d).

**Contribution of different mTORC1 targets to β-cell mass**. Analysis of β-cell mass at 5 months showed that reconstitution of S6K activity, 4EBP1/eIF4E or 4E-BP2/eIF4E signalling increased β-cell mass in *βraKO* mice (Fig. 4e and Supplementary Fig. 12). However, β-cell mass in *βraKO;caS6K* and *βraKO;Eif4ebp2^{−/−}* was higher than that observed in *βraKO;Eif4ebp1^{−/−}* mice (Fig. 4e and Supplementary Fig. 12). Surprisingly, restoration of S6K together with 4E-BP2/eIF4E signalling was sufficient to normalize β-cell mass but not hyperglycaemia (Fig. 4b,e). Assessment of components that contribute to β-cell mass showed that β-cell proliferation was rescued only in genetic models with 4E-BP2 deletion (Fig. 4f). On the other hand, S6K activation rescues primarily β-cell apoptosis and size (Fig. 4g,h). The levels of LC3-II were restored to normal in *βraKO;caS6K* and *βraKO;caS6K;Eif4ebp2^{−/−}* mice (Supplementary Fig. 13a). Surprisingly, this effect was not mediated by ULK phosphorylation since no changes were observed in islets of *caS6K* (101.5 ± 9.611 versus 80.9 ± 15.53 p-ULK/total ULK, $n = 4$, $P = 0.3143$). To explore the mechanisms linking S6K activity and autophagy, we focused on previous evidence showing that mTORC1/S6K signalling inhibits autophagosome formation by increasing the

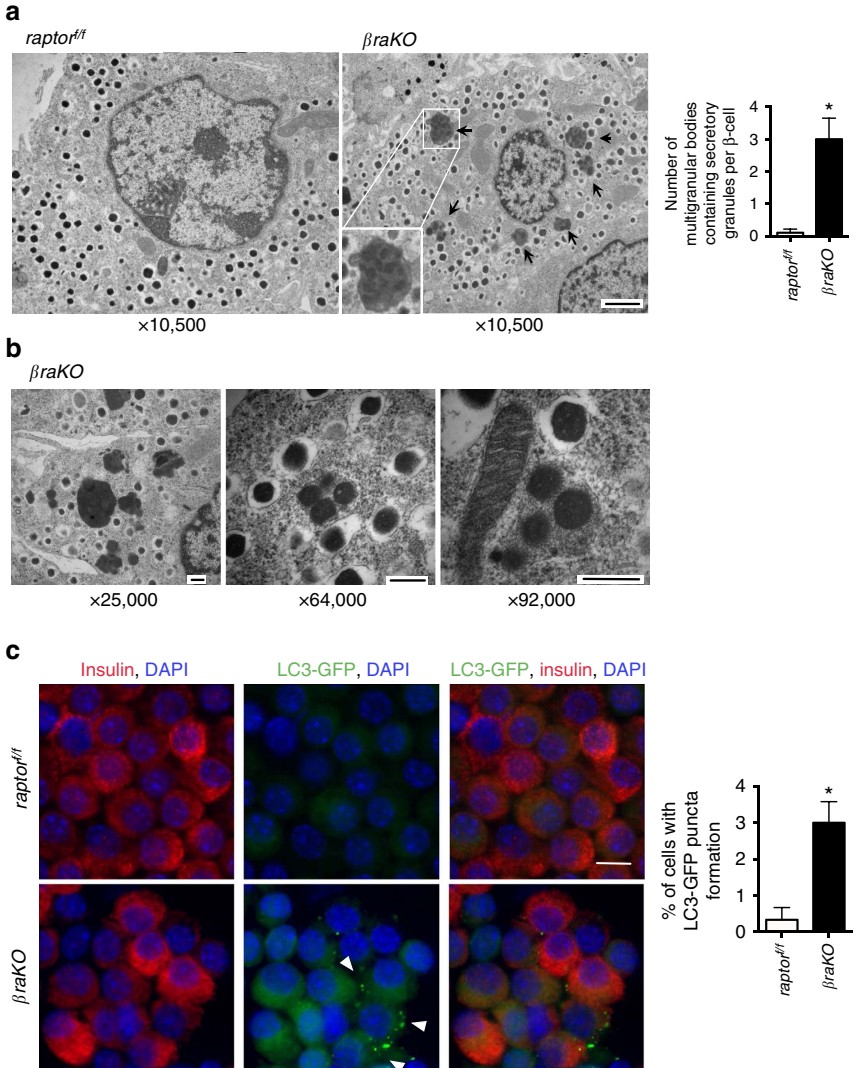

**Figure 2 | Assessment of autophagy in mice with Raptor deletion in β-cells. (a)** Electron microscopy images (left) of islets from *raptor^{f/f}* and *βraKO* mice at 30 days of age. Lysosomes containing insulin granules (crinophagy; arrows) are observed in β-cells from *βraKO* mice (inset shows magnification of crinophagy). Magnification: ×10,500. Images are representative of islets from three animals per group. Quantification of the number of multigranular bodies containing secretory granules per cell (right). **(b)** Electron microscopy images of multigranular bodies containing insulin secretory granules in *βraKO* mice at different magnifications. Scale bar, 1 μM in (**a**) and 300 nM in **b**. **(c)** Immunostaining for insulin (red), LC3-GFP (green) and DAPI (blue) in dispersed islets from *raptor^{f/f}* and *βraKO* mice at 40 days of age. Scale bar, 10 μM. Percentage of cells with LC3-GFP puncta formation (arrows) in dispersed cells from LC3-GFP (*raptor^{f/f}* and *βraKO*) mice at 40 days (n = 4). Data are shown as means ± s.e.m., *P < 0.05; nonparametric U-test (Mann–Whitney).

antiapoptotic protein FLIP$_L$ (refs 23,24). Indeed, overactivation of S6K induces FLIP$_L$ protein expression in islets (Supplementary Fig. 13b). In summary, we report that restoring 4E-BP2/eIF4E and S6K, but not 4E-BP1/eIF4E, has similar effects in improving fed glucose levels and β-cell mass in mice with inactivation of mTORC1 in β-cells. However, while 4E-BP2/eIF4E signalling is critical for β-cell proliferation downstream of mTORC1, S6K pathway controls cell size and apoptosis as well as regulation of autophagy.

**mTORC1 regulates insulin processing through 4E-BP2/eIF4E.** The previous results indicate that activation of S6K and 4E-BP2/eIF4E signalling in *βraKO* mice rescued the β-cell mass phenotype but failed to completely normalize fed glucose levels (Fig. 4a,b,e). To explore this further, we decided to assess the contribution of insulin processing by measuring insulin and proinsulin levels in all genetic mouse models. Circulating levels of proinsulin and proinsulin/insulin ratio were higher in *βraKO*

mice (Fig. 5a,c) strengthening the hypothesis that mTORC1 regulates insulin processing. A defect in insulin processing was confirmed by elevated immunostaining for proinsulin concomitantly with decreased insulin staining in β-cells of *βraKO* mice before the onset of hyperglycaemia (30 days of age; Fig. 5d). Lower insulin content was further confirmed in single β-cells by flow cytometry (Fig. 5e). Similar to *βraKO*, *MIP-βraKO^{f/f}* mice also displayed higher levels of proinsulin further supporting the role of mTORC1 in insulin processing (Fig. 5f). Interestingly, concomitant activation of 4E-BP2/eIF4E and S6K pathways in *βraKO* mice rescued serum insulin levels but only 4E-BP2/eIF4E signalling restored proinsulin levels and proinsulin/insulin ratio (Fig. 5a–c). In isolated islets, the proinsulin/insulin ratio was normalized in the *βraKO;Eif4ebp2^{−/−}*, but not in the *βraKO;caS6K;Eif4ebp2^{−/−}* mice (Fig. 5g and Supplementary Fig. 14a), suggesting that S6K overactivation neutralizes the beneficial effect of 4E-BP2/eIF4E activation on proinsulin processing.

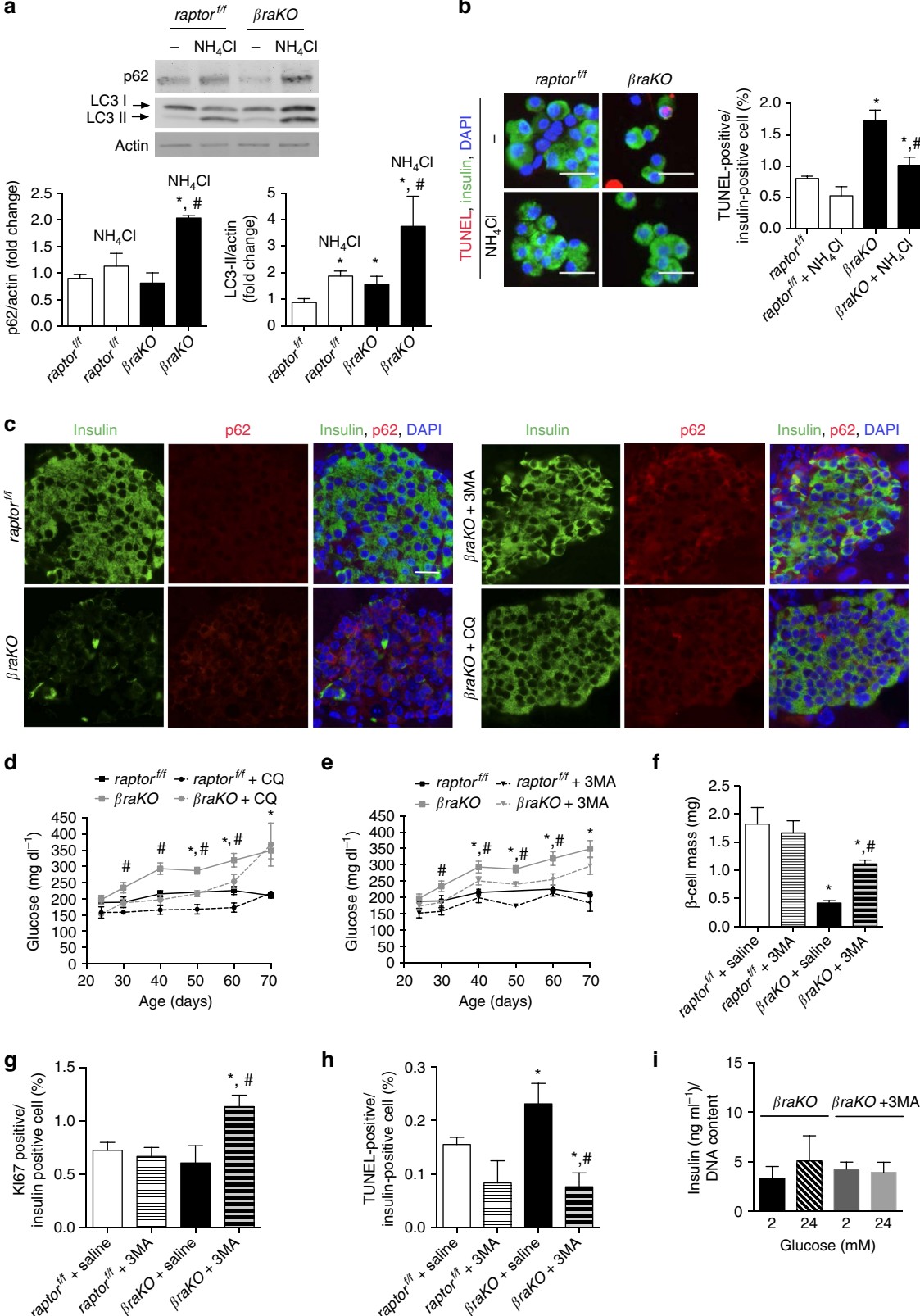

**S6K and 4E-BP2 converge on insulin signalling to regulate CPE.** To understand how mTORC1 controls insulin processing, we first assessed CPE levels in *βraKO* mice and reconstitution models. CPE expression was reduced in islets from *βraKO*, and control mice treated with the mTORC1 inhibitor rapamycin (Fig. 6a,b and Supplementary Fig. 14b,c). CPE was rescued by activation of 4E-BP2/eIF4E pathway in the *βraKO* mice but blunted when over-activation of S6K was present (Fig. 6a and Supplementary Fig. 15). In fact, islets from *caS6K* mice (expressing a rapamycin-resistant mutant[26]) exhibited reduced levels of CPE with or without

rapamycin treatment (Fig. 6b and Supplementary Fig. 14c). These results further support the crosstalk between S6K overactivation and 4E-BP2/eIF4E pathway on insulin processing. Recent studies demonstrated that CPE levels are controlled by an insulin/PDX1/eIF4G axis[29]. Knowing that S6K activation reduces insulin signalling by decreasing IRS1/2 levels[26] we hypothesized that S6K overactivation could lower CPE translation by a decrease in this previously described axis. To assess this, we evaluated the upstream components responsible for regulation of eIF4G levels by assessing the IRS2/PDX1/eIF4G1 axis in $Eif4bp2^{-/-}$, caS6K and caS6K;$Eif4bp2^{-/-}$ mice. First, we observed that IRS2 and PDX1 protein levels were diminished in the caS6K mice[25] (Fig. 6c), contrasting to the previously described higher levels in $Eif4bp2^{-/-}$ mice[27] (Fig. 6c). Interestingly, IRS2 and PDX1 were also reduced in the double-mutant caS6K;$Eif4bp2^{-/-}$ indicating that S6K overexpression neutralizes 4E-BP2 deletion effects on these proteins (Fig. 6c). Consistently, eIF4G protein levels were decreased in the βraKO;caS6K;$Eif4ebp2^{-/-}$ mice but normal in the βraKO;$Eif4ebp2^{-/-}$ (Fig. 6c) indicating that abnormalities in proinsulin processing in βraKO;caS6K;$Eif4ebp2^{-/-}$ mice were caused by reduced eIF4G and CPE levels. Finally, given the known role of CPE in β-cell apoptosis, we reconstituted CPE levels in MIP-βraKO[f/f] islets (4 weeks after TMX injection) by transduction with control or CPE-GFP adenoviruses (Ad. CPE-GFP[+]). Proinsulin levels were reduced in MIP-βraKO[f/f] GFP[+] β-cells but reconstitution of CPE failed to rescue apoptosis in MIP-βraKO[f/f] GFP[+] cells indicating that apoptosis was not directly caused by a reduction in CPE (Supplementary Fig. 16a–c).

Assessment of other hormone convertases critical for insulin processing[30–34] revealed that PC1/3 was reduced by 20% in βraKO mice and the levels were not rescued in βraKO;$Eif4ebp2^{-/-}$ or βraKO;caS6K;$Eif4ebp2^{-/-}$ (Supplementary Fig. 17a). PC2 levels were unchanged in the different experimental groups (Supplementary Fig. 17a). Rapamycin treatment of MIN6 cells decreased CPE but did not change PC1/3 levels (Supplementary Fig. 17b). Finally, MIN6 cells with stable silencing of 4E-BP2 (sh4ebp2) exhibited a 75% reduction in 4E-BP2 ($P < 0.05$, $n = 4$) and a twofold rise in CPE protein levels (Fig. 7a and Supplementary Fig. 18). However, CPE mRNA levels in sh4ebp2 cells (Fig. 7b) were unchanged, suggesting that the rise in CPE was post-transcriptional.

### 4E-BP2 regulates CPE levels by controlling mRNA translation.

The finding that CPE protein but not mRNA is regulated by mTORC1/4E-BP2/eIF4E pathway prompted us to investigate cap-dependent translation mechanisms. Removal of 4E-BP2 releases eIF4E, allowing eIF4E to interact with eIF4G and assembly of the eIF4F complex, resulting in enhanced cap-dependent translation[35]. The eIF4G:eIF4E interaction is important to drive translation of a subset of mRNAs with complex secondary structure in the 5′-UTR (characterized by high guanine cytosine (GC) content and thermodynamically stable structures (low $\Delta G$))[36]. Indeed, a complex secondary structure of the CPE 5′-UTR was supported by a more G/C rich (%GC 71.4) than average (actin) and lower free energy ($\Delta G$: −33.40 kcal mol[−1]), suggesting that this mRNA could be favourably translated by eIF4E (Supplementary Fig. 19)[37]. To test this hypothesis, we used 4E1RCat, an inhibitor of cap-dependent translation that prevents eIF4G:eIF4E interaction. Interestingly, 4E1RCat slightly reduced CPE levels in control cells (shCtrl) and dampened the increase in CPE levels observed in sh4ebp2 MIN6 cells (Fig. 7a). Examination of the polyribosome profile showed that sh4ebp2 MIN6 cells displayed an increase in the fraction of RNAs associated with polyribosomes (Fig. 7c, red line). Also, CPE mRNA levels in polyribosomal fractions were significantly increased in the polyribosomal fractions from sh4ebp2 MIN6 cells, evidencing that silencing 4E-BP2 enhances CPE translation (Fig. 7d). Interestingly, inhibition of eIF4E:eIF4G-dependent translation in AtT20 cells (a pituitary cell line that expresses CPE and proopiomelanocortin (POMC)) resulted in decreased CPE levels (Fig. 7e). Also, 4E1RCat treatment resulted in accumulation of POMC and a decrease in melanocortin (αMSH) levels, pointing to an impaired POMC processing and suggesting that this mechanism is not limited to β-cells (Fig. 7f). Therefore, we provide evidence supporting that the mTORC1/4E-BP2/eIF4E pathway regulates CPE protein level by increasing mRNA translation.

### Rapamycin recapitulates changes in insulin processing.

To confirm that inhibition of mTORC1 *in vivo* by rapamycin treatment recapitulates the defects in insulin processing observed in βraKO mice, we measured proinsulin in control mice treated with rapamycin intraperitoneally. Rapamycin treatment for 1 week was sufficient to increase proinsulin levels and alter proinsulin/insulin ratio in plasma and islets from control mice (Fig. 8a–c). More importantly, rapamycin treatment failed to increase plasma proinsulin levels, proinsulin/insulin ratio and proinsulin content in islets from $Eif4ebp2^{-/-}$ mice (Fig. 8a–c and Supplementary Fig. 20a). In addition, rapamycin reduced CPE expression in control islets but not in islets from $Eif4ebp2^{-/-}$ mice (Fig. 8d and Supplementary Fig. 20b). Finally, treatment of human islets with rapamycin for 48 h was sufficient to decrease CPE protein levels (Fig. 8e,f and Supplementary Fig. 20c). We were unable to show alterations in proinsulin:insulin ratio after prolonged treatment with rapamycin, as this was associated with significant toxicity. Disrupting eIF4E:eIF4G complex by 4E1RCat treatment also reduced CPE levels in human islets (Fig. 8g and Supplementary Fig. 20d). Thus, the experiments performed in human islets recapitulates the defects CPE protein expression.

### Discussion

Here we provide the first *in vivo* genetic dissection of mTORC1 signalling in insulin-sensitive tissues. These studies provide novel insights into how this pathway regulates different biological

**Figure 3 | Effect of *in vitro* and *in vivo* inhibition of autophagy in βraKO mice.** (**a**) Immunoblotting (upper) and quantification (lower) for p62, LC3-I/II and actin in islets from βraKO and raptor[f/f] (30–40 days of age) treated with or without NH₄Cl (20 mM) for 24 h. A representative image from four independent experiments is included and each lane shows the expression levels from one mouse. (**b**) Assessment of apoptosis after inhibition of autophagy in islets from raptor[f/f] and βraKO mice (30–40 days of age). Staining (left) and quantification (right) of TUNEL-/insulin-positive cells in dispersed islets from raptor[f/f] and βraKO treated or not with NH₄Cl for 24 h. Scale bar, 10 μM (n = 4). (**c**) Immunostaining for insulin (green), p62 (red) and DAPI (blue) in sections from 80-day-old raptor[f/f] and βraKO mice treated for 8 weeks with 3MA (15 mg kg[−1] in 0.9% saline), CQ (7 mg kg[−1] in 0.9% saline) or vehicle (saline). Treatment started in 18-day-old mice. Scale bar, 20 μM. (**d**) Random fed blood glucose levels in raptor[f/f] (control) and βraKO mice treated with control vehicle or CQ. (**e**) Random fed blood glucose levels in control (raptor[f/f]) and βraKO mice treated with control vehicle or 3MA. Same glucose values for control and βraKO mice are included in Fig. 2c,d. (**f**) Assessment of β-cell mass at 80 days of age in raptor[f/f] and βraKO mice treated with 3MA or control vehicle. (**g,h**) Assessments of β-cell proliferation and TUNEL in 80-day-old mice (n = 6; Fig. 5c–h). (**i**) GSIS determined by static incubation of isolated islets from βraKO mice treated with 3MA or control vehicle (n = 4). Data are shown as means ± s.e.m., *$P < 0.05$ versus raptor[f/f] (control) mice treated with saline, and #$P < 0.05$ versus βraKO mice treated with saline; nonparametric *U*-test (Mann–Whitney).

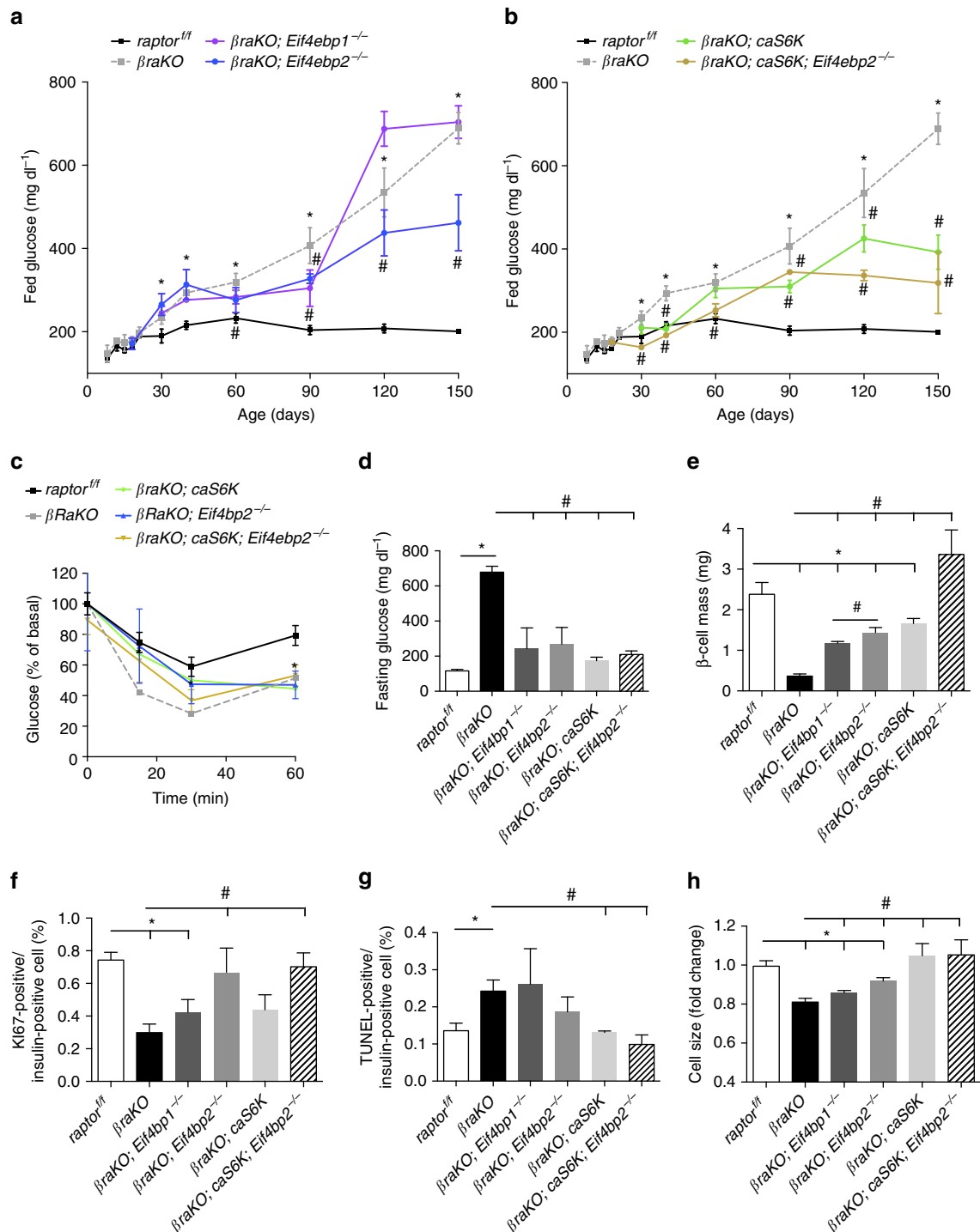

**Figure 4 | Glucose homeostasis and pancreas morphology in mice with genetic reconstitution of mTORC1 downstream targets.** (**a**) Random fed blood glucose levels in control (*raptor*$^{f/f}$), *βraKO* and *βraKO* mice crossed to 4E-BP1$^{-/-}$ (*βraKO; Eif4ebp1*$^{-/-}$) and 4E-BP2$^{-/-}$ mice (*βraKO; Eif4ebp2*$^{-/-}$). (**b**) Random fed glucose levels over time in control (*raptor*$^{f/f}$), *βraKO*, *βraKO* mice crossed to mice overexpressing a constitutively active S6K (*βraKO; cas6K*) and *βraKO* mice crossed to *cas6K* and 4E-BP2$^{-/-}$ mice (*βraKO; caS6K; Eif4ebp2*$^{-/-}$). Same glucose values for control and *βraKO* mice are included in **a** and **b**. (**c**) Insulin tolerance test at 90 days in the same group of mice. (**d**) Fasting glucose in 150 days of age. (**e–g**) Assessments of β-cell mass, β-cell proliferation and TUNEL at 150 days of age. (**h**) Measurement of cell size expressed as a fold change (*n* = 6; a–h). Data are shown as means ± s.e.m., *n* = 6 per group. \**P* < 0.05 versus *raptor*$^{f/f}$ (control) mice and #*P* < 0.05 versus *βraKO* mice; nonparametric *U*-test (Mann–Whitney).

processes in β-cells. We show that *raptor* inactivation in β-cells causes β-cell failure and genetic reconstitution of downstream components in these mice determined the contribution of different mTORC1 targets to β-cell loss in these mice. The genetic reconstitution studies in *βraKO* mice demonstrated that

downstream of mTORC1 (Supplementary Fig. 21a) (1) ULK1 and S6K axis control β-cell survival, (2) S6K mediates the effects on β-cell size and (3) 4E-BP2/eIF4E (and not 4E-BP1) induces β-cell proliferation. The studies herein also identified a novel mechanism for mTORC1-dependent translational control of insulin

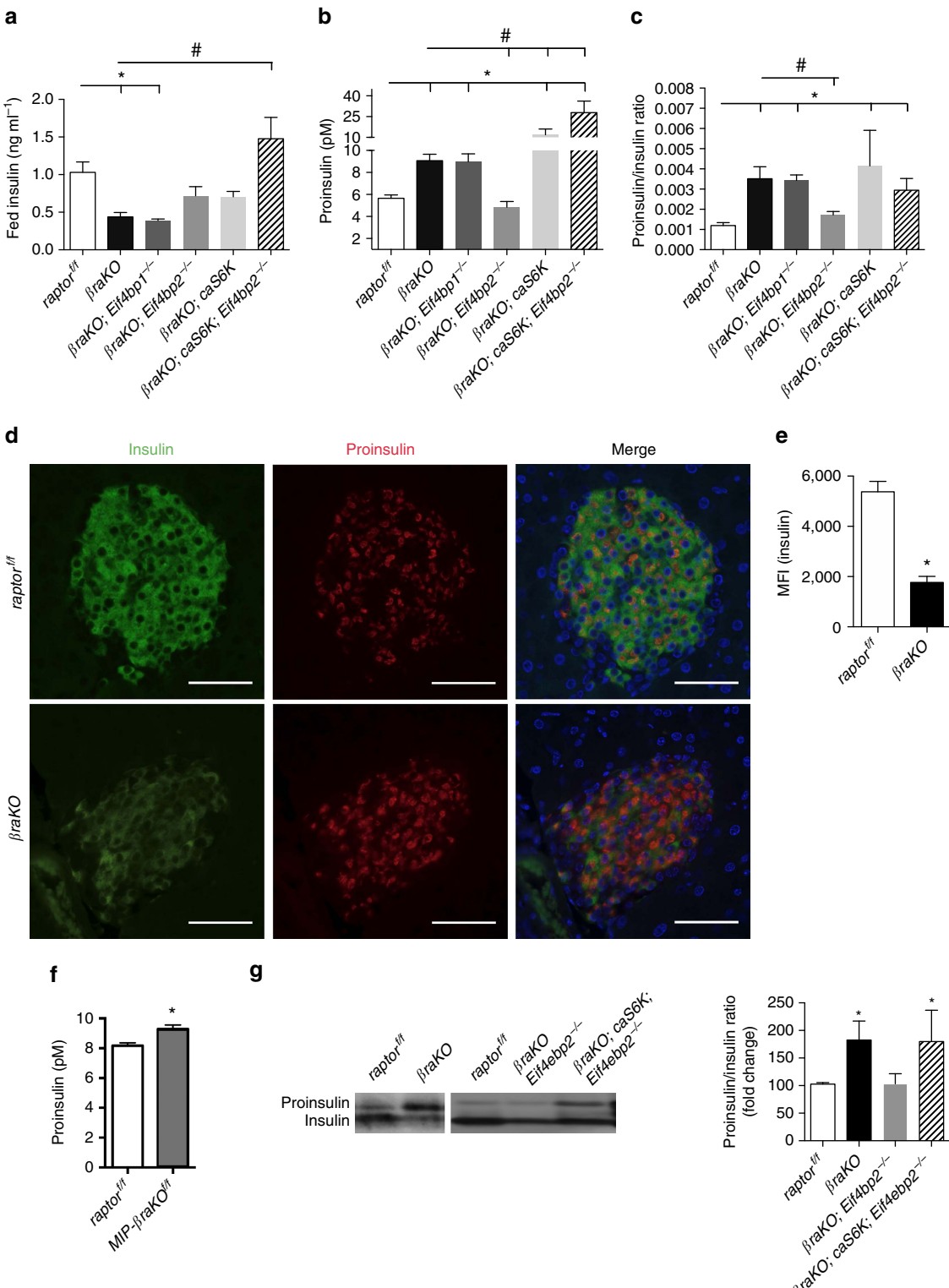

**Figure 5 | Assessment of insulin level in different genetic models uncovers a role for mTORC1 on insulin processing.** (**a**) Random fed serum insulin levels in *raptor*$^{f/f}$, *βraKO*, *βraKO; Eif4ebp1*$^{-/-}$, *βraKO; Eif4ebp2*$^{-/-}$, *βraKO; cas6K* and *βraKO; caS6K;Eif4ebp2*$^{-/-}$ mice at 150 days of age. (**b,c**) Random fed serum proinsulin and proinsulin/insulin ratio in 150-day-old mice ($n = 6$; **a–c**). (**d**) Immunostaining for proinsulin (red), insulin (green) and DAPI (blue) in pancreatic sections from 30-day-old *raptor*$^{f/f}$ and *βraKO* mice. Scale bar, 50 µM. Images are representative of three different mice. (**e**) Insulin content per β-cell measured by flow cytometry and expressed as mean fluorescent intensity (MFI; $n = 4$; 30–40 days of age). (**f**) Random fed serum proinsulin levels measured by ELISA in control and *MIP-βraKO*$^{f/f}$ mice at 8 weeks after TMX injection ($n = 5$). (**g**) Immunoblotting and quantification of insulin and proinsulin in islets from 30-day-old *βraKO*, *βraKO; Eif4ebp2*$^{-/-}$, *βraKO;caS6K; Eif4ebp2*$^{-/-}$ and control mice ($n = 4$). Data are shown as means ± s.e.m., $n = 6$ per group. *$P < 0.05$ versus *raptor*$^{f/f}$ (control) mice and #$P < 0.05$ versus *βraKO* mice; nonparametric *U*-test (Mann–Whitney).

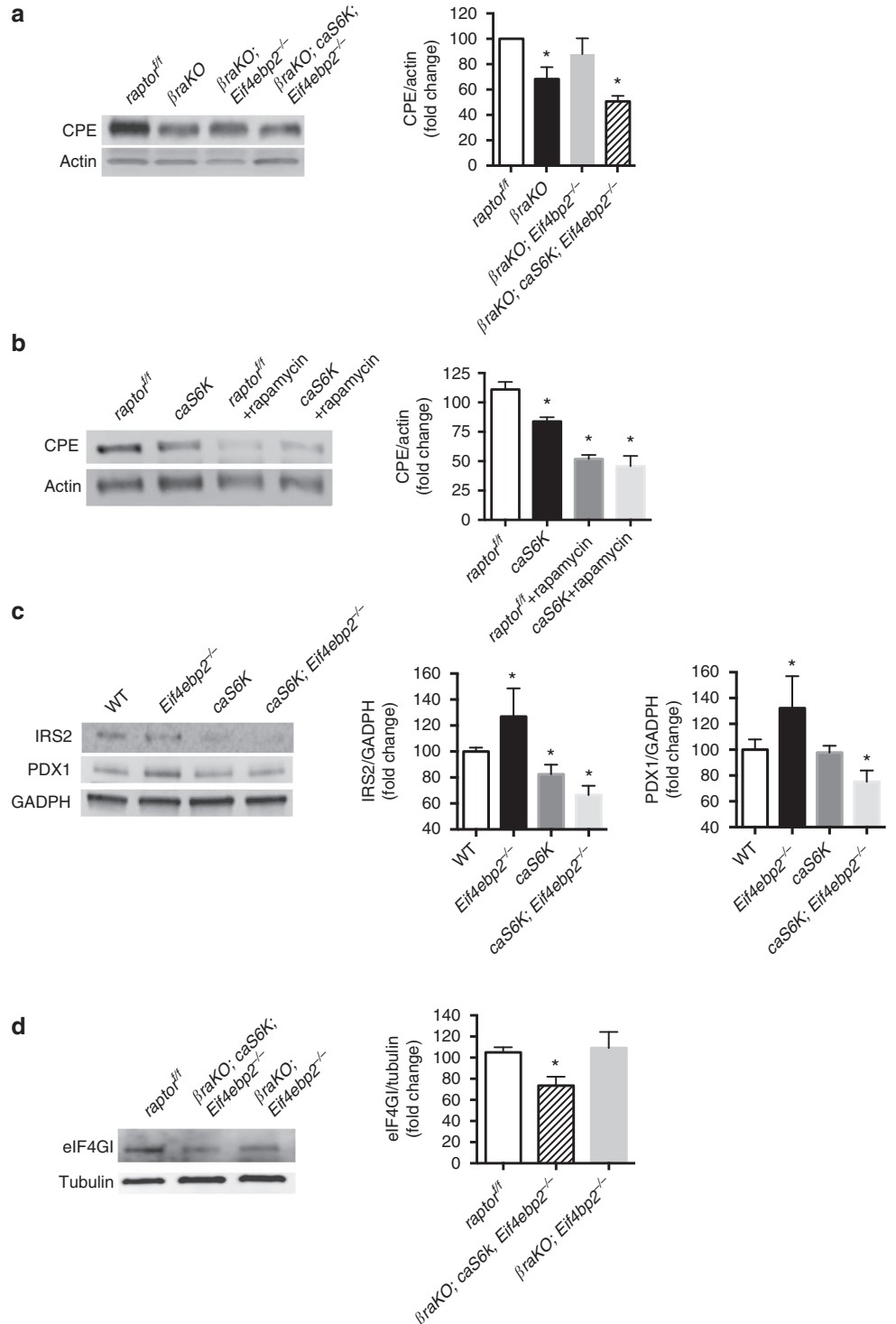

**Figure 6 | mTORC1 controls CPE levels via a crosstalk between S6K and 4E-BP2/eIF4E pathway.** (**a**) Immunoblotting and quantification of CPE and actin in islets from 30-day-old *βraKO*, *βraKO; Eif4ebp2*$^{-/-}$, *βraKO; caS6K; Eif4ebp2*$^{-/-}$ and control mice (*n* = 4). (**b**) Immunoblotting and quantification of CPE and actin in islets from 90-day-old transgenic mice overexpressing a constitutively active S6K (*caS6K*) and control mice treated with vehicle (DMSO) or rapamycin (30 nM) *in vitro* for 24 h (*n* = 4). (**c**) Immunoblotting and quantifications for IRS2, PDX1 and GADPH in wild-type, *Eif4ebp2*$^{-/-}$, *caS6K* and *caS6K; Eif4ebp2*$^{-/-}$ islets. A representative image from four independent experiments is included and each lane shows the expression levels from one mouse (*n* = 4). (**d**) Immunoblotting for eIF4GI and tubulin in *βraKO; Eif4ebp2*$^{-/-}$, *βraKO; caS6K; Eif4ebp2*$^{-/-}$ and control mice (*n* = 4). Data are shown as means ± s.e.m., \**P* < 0.05; nonparametric *U*-test (Mann–Whitney).

processing in a 4E-BP2/eIF4E-dependent manner by regulating cap-dependent translation of CPE. The decrease in CPE levels after mTORC1 inhibition by rapamycin treatment of human islets suggests that this mechanism could be relevant to humans.

The development of hyperglycaemia in *βraKO* mice resulted in part from a progressive reduction in β-cell mass due to decreased proliferation, reduced cell size and increased apoptosis. In contrast to the severe phenotype of *βraKO* mice, β-cell mass in

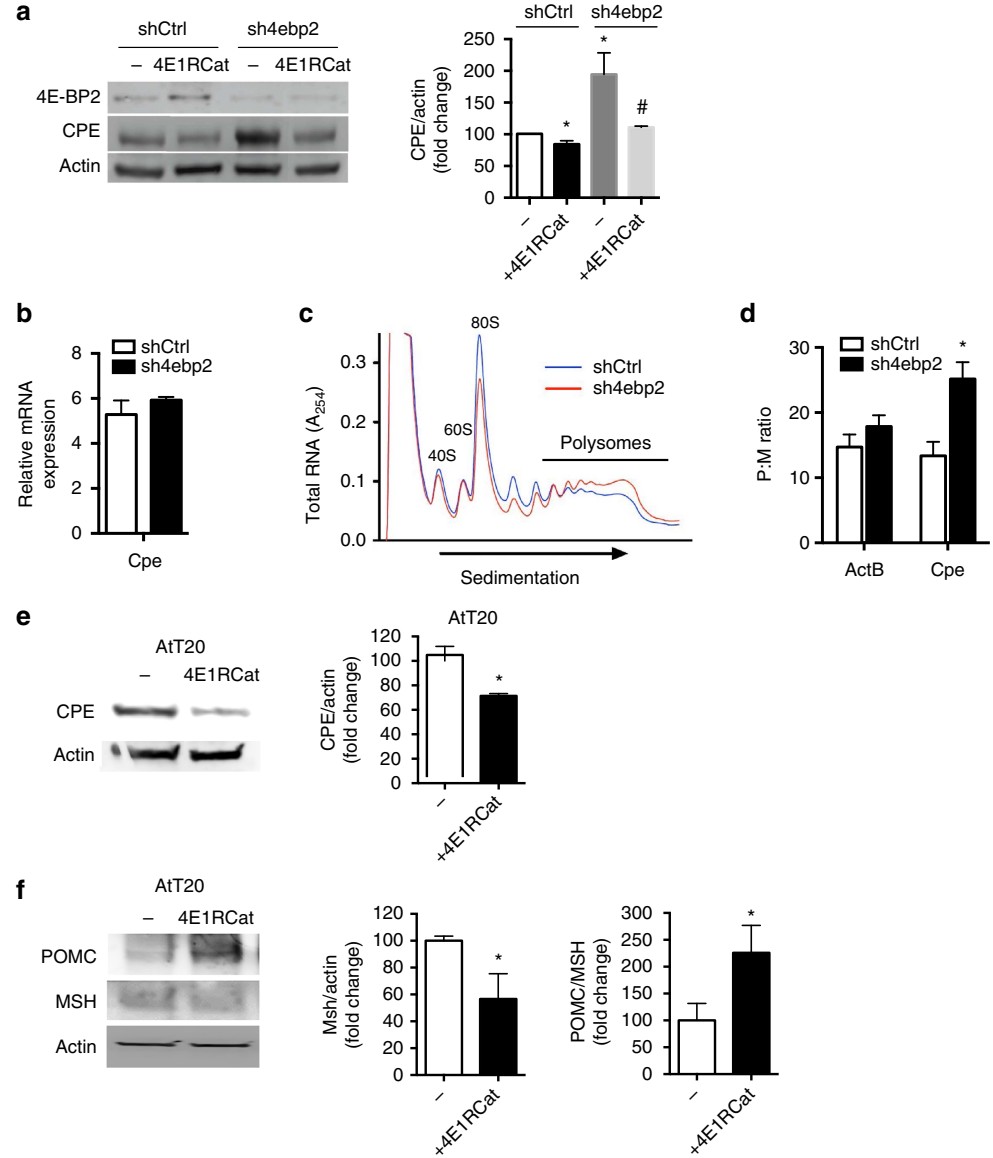

**Figure 7 | 4E-BP2 controls CPE levels by modulation of cap-dependent translation.** (**a**) Immunoblotting and quantification of CPE and actin in 4E-BP2 knockdown (sh4ebp2) and control MIN6 cells (shCtrl) treated or not with 4E1RCat (25 μm) for 24 h (n = 6). (**b**) CPE expression at the mRNA level by RT–PCR in control and in sh4ebp2 cells (n = 4). (**c**) Polyribosome profile of control and sh4ebp2 cells cultured in regular conditions (n = 4). (**d**) CPE mRNA levels in polysomal and monosomal fractions from control and sh4ebp2 cells cultured in normal conditions. Data expressed as a ratio of CPE expression in polysome/monosome (P/M) fractions (n = 4). (**e**) Immunoblotting and quantification of CPE and actin in AtT20 cells treated or not with 4E1RCat for 24 h (n = 4). (**f**) Immunoblotting and quantification of POMC, αMSH and actin in AtT20 cells treated or not with 4E1RCat for 24 h (n = 4). Data are shown as means ± s.e.m., *P < 0.05; nonparametric U-test (Mann–Whitney).

*MIP-βraKO^{f/f}* mice was maintained after 8 weeks of TMX suggesting that Raptor plays more important roles in early stages of postnatal development and β-cell maturation. The lack of changes in β-cell mass in *MIP-βraKO^{f/f}* mice allowed us to identify an important role of mTORC1 on insulin secretion, and preliminary studies showed that *MIP-βraKO^{f/f}* mice exhibit reduced intracellular calcium after glucose stimulation (unpublished data). This suggests that events proximal to calcium influx are involved but it is possible that other steps contribute to this phenotype. The use of *MIP-CreERT* mice has some limitations due to the expression of the human growth hormone (hGH) minigene. While we cannot completely exclude the possibility that hGH expression from the *MIP-CreERT* transgene ameliorates the loss of β-cell mass in *MIP-βraKO^{f/f}* mice, we believe that it is unlikely that hGH expression contributes to the alterations in

glucose tolerance as *MIP-βraKO^{f/f}* mice injected with corn oil exhibited normal glucose tolerance. The partial restoration of β-cell mass and TUNEL in *βraKO* mice treated with autophagy inhibitors suggests that autophagy plays a role in the loss of β-cells (Fig. 3b,f,h). Further experiments crossing *βraKO* mice with genetic models of decreased autophagy could be performed to test this hypothesis.

The *βraKO* model showed that chronic autophagy is detrimental for β-cells. These results are in marked contrast with published experiments showing that induction of autophagy by rapamycin treatment improved diabetes, increased pancreatic insulin content and prevented β-cell apoptosis in Akita mice implying that increased autophagy is beneficial in this model of ER stress induced by misfolded insulin[38,39]. The divergence in survival outcomes in these two models of autophagy is unclear

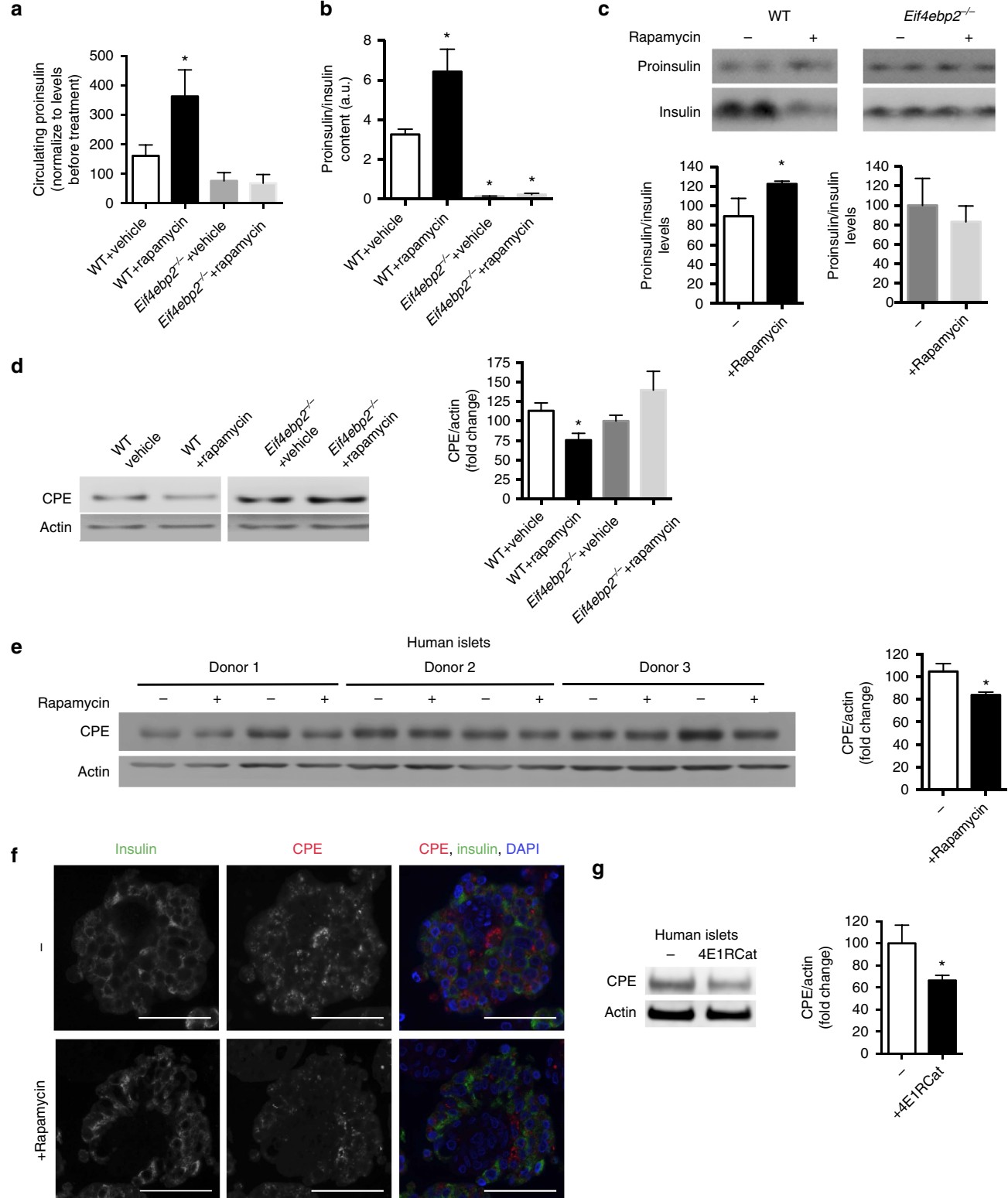

**Figure 8 | Rapamycin treatment impairs insulin processing and CPE levels in mice and human islets *in vitro*.** (**a**) Circulating proinsulin in serum from mice treated with intraperitoneal injections of rapamycin or control vehicle (2% ethanol) every other day for 9 days (treatment started at 3 months of age). (**b**) Proinsulin:insulin levels measured by ELISA in islets obtained from mice treated with rapamycin as described in **a**. (**c**) Immunoblotting and quantitative band densitometry of proinsulin and insulin in islets obtained from control and *Eif4ebp2*$^{-/-}$ mice treated with rapamycin or control vehicle (treatment started at 3 months of age). (**d**) Immunoblotting and quantification of CPE and actin in islets from *Eif4ebp2*$^{-/-}$ and control mice treated with control vehicle and rapamycin ($n = 4$ in **a**–**d**). (**e**) Immunoblotting and quantification of CPE and actin in human islets from different donors treated with vehicle control or rapamycin (30 nm) for 48 h ($n = 4$ for quantification). (**f**) Staining for CPE (red), insulin (green) and DAPI (blue) in human islets treated with rapamycin (30 nM) or control vehicle (DMSO) for 48 h. Scale bars, 50 μM. Data are representative of three per group. (**g**) Immunoblotting and quantification of CPE and actin in human islets treated with 41ERCat (25 μm) or vehicle (DMSO) for 24 h ($n = 4$). Data are shown as means ± s.e.m., *$P < 0.05$; nonparametric *U*-test (Mann–Whitney).

but it is possible that the induction of autophagy by different mechanisms could play a role (mutant insulin/ER stress versus genetic starvation by inactivation of mTORC1). In addition, it is unlikely that rapamycin treatment replicates all the biological effects of *raptor* deletion. Finally, other published data have demonstrated that inhibition of autophagy in *Atg7*$^{-/-}$ mice results in maintaining the structure, mass and function of pancreatic β-cells[40,41]. Taken together, these data support the concept that optimal levels of autophagy are essential for proper β-cell function and those conditions of extreme upregulation or inhibition of autophagy can negatively impact on β-cells mass.

Our studies showed that *raptor* inactivation exhibits increased canonical macroautophagy. This was supported by reduced ULK1 phosphorylation, increases in p62 staining and augmented puncta and LC3-II levels in the basal state (Supplementary Fig. 8c, and Figs 2c and 3c). In addition, higher p62 staining and LC3-II levels after inhibition of autophagy in *βraKO* mice is consistent with the increase in autophagy flux as described[42]. We also found that secretory granules are massively delivered for lysosomal degradation by electron microscopy. However, we failed to see evidence of macroautophagy (double-membrane structures) by electron microscopy at 30 and 60 days of age. There are several potential explanations for this finding: (1) autophagosome flux is a very rapid process in β-cells and is rarely detected under steady-state conditions[40]; (2) macroautophagy may be less prominent in β-cells than in other tissues[43]; and (3) there are major differences in autophagic vacuoles formation between β-cells and other organs.

Genetic reconstitution experiments demonstrate that hyperglycaemia in *βraKO* mice was partially rescued by gain of 4E-BP2/eiF4E, S6K or both. It is unlikely that enhanced insulin sensitivity observed in the reconstitution experiments explain the metabolic differences in *βraKO*, *βraKO;caS6K*, *βraKO;Eif4ebp2*$^{-/-}$ and *βraKO;caS6K;Eif4ebp2*$^{-/-}$ groups. These studies suggest that downstream of mTORC1 (1) 4E-BP2/eIF4E activation induces β-cell proliferation, (2) 4E-BP2/eIF4E controls insulin processing by regulating CPE translation, (3) ULK1-mediated effects in autophagy and survival contribute to the decrease in β-cell mass in *βraKO* and (4) reconstitution of S6K activity improves cell size and β-cell survival. The potential effect of S6K on regulation of β-cell autophagy is interesting, and our experiments are consistent with a model in which mTORC1/S6K inhibits autophagy by increasing FLIP$_L$ levels and this process ultimately inhibits autophagy-mediated cell death by binding to ATG3 and decreasing LC3 (refs 44,45). These results are in marked contrast with published data showing that phosphorylation of S6K is considered as pro-autophagy[46–48]. The effect of S6K in autophagy is controversial[49] but discrepancies in the literature could be explained in part by the following: (1) different mechanisms and duration of mTOR inhibition (amino-acid deprivation/rapamycin versus genetic inactivation and chronic versus acute); (2) our S6K overactivation model provides a non-physiological state of permanent S6K activation; and (3) previous work showing that S6K promotes autophagy was obtained in a model of genetic deletion of mTOR in flies[48]. However, in this model mTOR inactivation disrupts both mTORC1 and mTORC2, and mTORC2/Akt-FoxO pathway may be also involved in the control of autophagy[50]. More studies should be designed to clarify the role of S6K in autophagy in β-cells and other tissues. The current studies identified a role for mTORC1 on proinsulin processing due to a reduction in CPE and PC1/3 levels. Our findings showed that rescue of the insulin processing defect in *βraKO* mice was achieved by reconstitution of 4E-BP2/eIF4E signalling. This finding is consistent with a novel role of mTORC1/4E-BP2/eIF4E axis on regulation of insulin processing by controlling CPE expression. The complex secondary structure of the CPE 5′-UTR and the higher CPE mRNA levels in polyribosomal fractions in sh4ebp2 cells indicate that 4E-BP2/eIF4E regulates CPE translation, and data using the 4E1RCat further support this conclusion (Fig. 7a–d). Interestingly, normalization of insulin processing in *βraKO;Eif4ebp2*$^{-/-}$ disappeared when S6K was concomitantly overexpressed in *βraKO;caS6K;Eif4ebp2*$^{-/-}$, suggesting that 4E-BP2 and S6K converge on a common pathway to regulate CPE translation and that the negative effect of S6K overactivation on CPE is likely dominant. Published studies and the current experiments are consistent with a model in which insulin signalling controls insulin processing and CPE translation by two parallel pathways that converge on the formation of the translation initiation complex eIF4E:eIF4G: (1) a previously reported pathway controlled by insulin signalling IRS1/2/PDX1 that regulates eIF4G levels (Fig. 6c,d and Supplementary Fig. 21a,b). Our studies showed that this axis is disrupted by the negative feedback on IRS2 signalling induced by S6K overactivation. (2) A second pathway mediated by 4E-BP2 controlling the availability of eIF4E to form the eIF4E:eIF4G complex (Supplementary Fig. 21b). In this model, S6K overactivation neutralizes the effect of 4E-BP2 on CPE and proinsulin processing by reducing IRS2/PDX1 and ultimately eIF4G levels, a limiting partner for the formation of eIF4EG:eIF4E complex and regulation of CPE translation (Fig. 6c,d). The interaction between S6K/IRS2/PDX1/eIF4G and 4E-BP2/eIF4E pathways to control insulin processing during normal states or insulin resistance is unclear but we speculate prolonged overactivation of mTORC1 in conditions of chronic overnutrition results in a predominant S6K negative-feedback loop with decrease in IRS1/2 levels and reduction of IRS/PDX1/eIF4G/CPE pathway. This could explain the increase in proinsulin observed in early stages of type 2 diabetes. Finally, our data in mice and human islets treated with rapamycin indicate that the effect of mTORC1 on CPE and insulin processing operate *in vivo* and further confirm that mTORC1 regulates CPE levels in humans (Fig. 8e–g). Finally, these studies also showed that PC1/3 levels were reduced in *βraKO* mice and could play a role in the insulin-processing defect. How mTORC1 regulates PC1/3 is unclear and more experiments need to be done to study the role of PC1/3 in *βraKO* β-cells.

In summary, these results show that β-cell mTORC1 signalling controls glucose homeostasis by regulating β-cell mass, proliferation, apoptosis and insulin secretion. These findings also reveal a novel mechanism for mTORC1 in controlling insulin processing by regulating CPE translation in a 4E-BP2/eIF4E-dependent manner. This mechanism of hormone-processing regulation provides a new mechanism to explain the higher incidence of diabetes after rapamycin treatment in humans. As such, mTORC1 inhibition could have implications in the regulation of many biological processes that require CPE-mediated peptide processing, including thyroid hormone homeostasis, appetite control and obesity, fertility and neurodegeneration. Finally, it is possible that modulation of mTORC1/4E-BP2 axis could provide a useful therapeutic tool to improve the β-cell in pathological conditions.

## Methods

**Animals.** *RIP-Cre*, *MIP-CreERT* (*MIP-Cre*), global 4E-BP1 and two deficient mice (*Eif4ebp1* and *2*$^{-/-}$), *raptor*$^{fl/fl}$ and mice with transgenic overexpression of a rapamycin resistant constitutively active form of S6K in β-cells (*caS6K*) have been previously described[8,13,19,26,28,51]. B6.Cg-Gt(ROSA)26Sor$^{tm6(CAG-ZsGreen1)Hze}$/J (CAG-GFP) reporter transgenic animal mouse was purchased from The Jackson Laboratory (Stock number 007906). Studies were performed on mice on C57BL6J background. Results of the experiments are shown for male mice but phenotypes were validated in female mice (Supplementary Fig. 3a) at ages shown in figure legends. All animals were maintained on a 12 h light–dark cycle. We thank Dr N. Mizushima (University of Tokyo, Japan) for proving us the LC3-GFP mouse model. All procedures were performed in accordance with the University of Michigan Institutional Animal Care and Use Committee- and University of Miami-approved protocols.

**Metabolic studies.** Blood glucose levels were determined from blood obtained from the tail vein using ACCU-CHEK II glucometer (Roche). Fasting glucose and insulin were measured after overnight fasting. Glucose tolerance tests and GSIS were performed on overnight-fasted animals by injecting glucose intraperitoneally (2 and 3 mg kg$^{-1}$, respectively)[52]. Plasma insulin and proinsulin levels were determined using a Mouse Ultrasensitive Insulin ELISA kit and Mouse Proinsulin ELISA kit, respectively (ALPCO Immunoassays).

**Preparation and *in vivo* treatment with different agents.** TMX (Sigma) was dissolved in corn oil (Sigma) to a final concentration of 20 mg ml$^{-1}$. *raptor* deletion in mature β-cells was obtained by subcutaneous injections of 2 mg of TMX (corn oil in control mice) every day for 3 consecutive days in 2-month-old control and *MIP-Cre* mice. Systemic administration of 3MA (Cayman) was performed by daily intraperitoneal injections (15 mg kg$^{-1}$ in 0.9% saline) into 18-day-old *βraKO* and control littermates for 8 weeks. For CQ experiments, 18-day-old mice received weekly intraperitoneal injections of CQ diphosphate (7 mg kg$^{-1}$ and dissolved in 0.9% saline; Sigma) or saline for 8 weeks. For rapamycin (LC Laboratories) treatment *in vivo*, rapamycin was dissolved in 100% ethanol and stored at $-20\,°C$. Stock solution was further diluted in an aqueous solution of 5.2% Tween 80 and 5.2% PEG 400 (final ethanol concentration, 2%). Control and *Eif4ebp2*$^{-/-}$ mice were injected with rapamycin intraperitoneally (2 mg kg$^{-1}$ body weight) or vehicle (2% ethanol) every other day for 9 days. Islets from these mice were isolated and immediately lysed after isolation.

**Islets studies.** After islet isolation[52], islets were maintained at 37 °C in an atmosphere containing 20% oxygen and 5% $CO_2$. Insulin secretion from isolated islets was assessed by static incubation[52]. Briefly, after overnight culture in RPMI containing 5 mM glucose and 10% FBS, islets were pre-cultured for 1 h in Krebs–Ringer medium containing 2 mM glucose. Groups of 10 islets in triplicates were then incubated in Krebs–Ringer medium containing 2 or 24 mM glucose for 1 h. Secreted insulin in the supernatant and insulin content was then measured using Mouse Ultrasensitive Insulin ELISA kit (ALPCO Immunoassays) and normalized to DNA content. Experiments to assess the inhibition of autophagy *in vitro* were performed by culturing islets for 24 h in RPMI containing NH4Cl (20 mM; Sigma) and 5 mM glucose. For rapamycin treatment *in vitro*, islets were cultured in RPMI containing 5 mM glucose and rapamycin (30 nM; Sigma) for 24 or 48 h as indicated in the corresponding figure legends. Similar culture conditions were used for experiments with 4E1RCat (25 μm; Sigma), an inhibitor of the interaction between eIF4E:eIF4G. Human islets were obtained through the Integrated Islet Distribution Program (Supplementary Table 1).

**Western blotting.** Islets from an individual mouse (120–150 islets) were lysed in lysis buffer (125 mM Tris, pH 7, 2% SDS and 1 mM dithiothreitol) containing a protease inhibitor cocktail (Roche Diagnostics). Protein quantity was measured by a bicinchoninic acid assay method, and 40 μg of protein were loaded on SDS–PAGE gels and separated by electrophoresis. Separated proteins were transferred onto polyvinylidene difluoride membranes (Millipore, Bedford, MA) overnight. After blocking for 1 h in 1× Tris-buffered saline–1% Tween 20–5% milk, membranes were incubated overnight at 4 °C with a primary antibody diluted in 1× Tris-buffered saline–1% Tween 20–5% milk followed by 1 h incubation at room temperature with horseradish peroxidase-conjugated secondary antibodies. Antibodies used for immunoblotting are included in Supplementary Table 2, and membranes were developed using Western Bright Sirius Kit (BioExpress). Band densitometry was determined by measuring pixel intensity using NIH Image J software (v1.49d ref. 53) freely available at http://rsb.info.nih.gov/ij/index.html) and normalized to tubulin, actin or total protein in the same membrane. Images have been cropped for presentation. Full-size images for the most important western blots are presented in Supplementary Figs.

**Adenoviral infection.** After overnight culture in RPMI containing 5 mM glucose, islets were infected with adenoviruses carrying the cytomegalovirus promoter (Ad. CMV) or CPE and GFP under the control of the CMV promoter (Ad. CPE-GFP (ADV-256014), Vector BioLabs). The particle:plaque-forming unit ratio of the stock virus used in the experiments was 300. After 3 days of culture, islets were dispersed, fixed and stained for CPE, active caspase 3, proinsulin and insulin.

**Flow cytometry.** After overnight culture in RPMI containing 5 mM glucose, islets were dispersed into a single-cell suspension and fixed with BD Pharmingen Transcription Factor Phospho Buffer Set (BD Biosciences). Dispersed cells were incubated overnight with conjugated antibodies at 4 °C. Dead cells were excluded by Ghost Dye Red 780 (Tonbo), and signal intensity from single stained cells and GFP was analysed by mean fluorescent intensity in insulin-positive cells using BD LSR II (BD Biosciences). Antibodies used are included in Supplementary Table 2.

**Quantitative real-time PCR.** Total RNA was isolated using RNeasy (Qiagen) followed by cDNA synthesis using High Capacity cDNA Reverse Transcription Kit (Applied Biosystems) according to the manufacturer's protocol. Real-time PCR was performed on an ABI 7000 sequence detection system using POWER SYBR-Green

PCR Master MIX (Applied Biosystems). Primers were purchased from IDT Technologies. Primer pair for CPE was as follows: 5′-CGGACAAACCCTTTAA-CACC-3′ (forward); 5′-CCGGAAGAGACTCTCAAAAGC-3′ (reverse).

**Polyribosomal profiling and gradient fraction quantitative PCR.** For polyribosomal profiling studies, sh4ebp2 and control MIN6 cells[27] were lysed and processed on sucrose gradients[54]. A piston gradient fractionator (BioComp) was used to measure RNA $A_{254}$ with an in-line ultraviolet monitor, and gradients were collected in ten 1 ml fractions. Fractions 1–5 were combined as the monosome pool, and fractions 6–10 were combined as the polysome pool. The percentage of message recovered in the monosome and polysome pools was determined for each condition. mRNA for polysomal and monosomal fraction was isolated as described above.

**Immunofluorescence and morphometry.** Formalin-fixed pancreatic tissues were embedded in paraffin using standard techniques. Immunofluorescence staining was performed using primary antibodies described on Supplementary Table 1. Fluorescent images were acquired using a microscope (Leica DM5500B) with a motorized stage using a camera (Leica Microsystems, DFC360FX), interfaced with the OASIS-blue PCI controller and controlled by the Surveyor software (version 7.1, Objective Imaging Ltd). β-cell ratio assessment was calculated by measuring insulin and acinar areas using Image Pro Software (version 7 Media Cybernetics, Inc.) in five insulin-stained sections (5 μm) that were 200 μm apart. To calculate β-cell mass, β-cell to acinar ratio was then multiplied by the pancreas weight. Assessment of proliferation was performed in insulin- and Ki67-stained sections, and included at least 3,000 cells per animal. Apoptosis was determined using TUNEL assay (ApopTag Red In Situ Apoptosis Detection Kit, Chemicon) in insulin-stained sections. At least 3,000 β-cells were counted for each animal. Cell size was determined by immunostaining sections for β-catenin and measuring the areas of individual β-cells from different experimental groups using NIH Image J software (v1.49d (ref. 53) freely available at http://rsb.info.nih.gov/ij/index.html). For dispersed cell staining, islets were gently dispersed after 5 min incubation with trypsin–EDTA (0.25% trypsin and 1 mM EDTA) in Hanks' balanced salt solution without $Ca^{2+}$ and $Mg^{2+}$ (Gibco Invitrogen) at 37 °C followed by fixation in 4% methanol-free formaldehyde onto poly-L-lysine-coated slides. All the morphologic measurements were performed in blinded manner.

**Electron microscopy.** Ultrastructural characterization by transmission electron microscopy was performed after overnight culture (in RPMI containing 5 mM glucose at 37 °C). Islets were then fixed with 2% glutaraldehyde and then dehydrated and embedded in Epon by the Microscopy & Image Analysis Laboratory Core (MiCORES). Ultrathin sections were stained with uranyl acetate and lead citrate, and images were recorded digitally using a Philips CM-100 electron microscope.

**Statistical analysis.** The statistical significance of differences between the various conditions was determined by nonparametric $U$-test (Mann–Whitney) using Prism version 6.0d (GraphPad Software, San Diego, CA). Data are presented as mean ± s.e.m. and were considered statistically significant when the $P$ value was < 0.05.

**Data availability.** All relevant data are available from the authors on request.

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

## Acknowledgements

We thank Dr Peter Arvan for discussion of the data and Drs Corentin Cras-Méneur and Lynda Elghazi for technical assistance with newborn pancreas dissection. We acknowledge funding resources for this essential contribution to this work. E.B.-M. is supported by the National Institutes of Health (NIH) Grant RO1-DK073716, DK084236, MERIT award IBX002728A and Juvenile Diabetes Research Foundation (JDRF) grant 17-2013-416. We acknowledge support from the Morphology and Image Analysis Core, Metabolomics Core and Phenotyping Core from the Michigan Diabetes Research Center (MDRC) (P30 DK020572).

## Author contributions

M.B.-R., R.B., M.J.-P., N.B., M.H., R.G.M. and M.L. performed experiments and analysed results; N.S., M.A.R. and M.N.H. generated mice; M.B.-R. and E.B.-M. designed experiments; M.B.-R., J.P.W. and E.B.-M. wrote the article. All authors contributed to discussion and reviewed/edited manuscript.

## Additional information

**Competing interests:** The authors declare no competing financial interests.

