## [Peer review file · Nature Communications]

Reviewers' comments:

Reviewer #1 (Remarks to the Author):

The authors explore the effects of raptor knockout in beta cells (bRaKO). They describe a phenotype of hypoinsulinemia and hyperglycemia, associated with reduced beta cell mass. The authors suggest that autophagy has deleterious effects in for beta cell survival in bRaKO. They also analyze the contribution of mTORC1 targets, uncovering the specific role of 4ebp2 for CPE translation and insulin processing. The data are mostly clear, but I am confused by the conclusions: specifically, the authors' conclusions on autophagy are in contrast to existing literature and other models of mTOR gain-of-function. The second part seems to be a bit of an afterthought, and is not well integrated with the first. The work is well done, but the conclusions a bit fuzzy.

1. Figure 2B shows percentage of LC3-positive beta cells (in Lc3-gfp transgenic mice). LC3 should be expressed in all cells, but it appears to be detected only detected in few cells. Is there an explanation for this? LC3 expression alone is not sufficient to conclude that there is autophagy. It would be useful to obtain higher magnification images to monitor subcellular distribution of LC3 (diffuse in cytoplasm vs. accumulation in puncta) which is characteristic of LC3-I and LC3-II respectively (the latter is representative of ongoing autophagy, even though a single measurement doesn't provide information on rates of autophagy).

2. Figure 2C. The authors show increased numbers of insulin granule-containing lysosomes (crinophagy). Generally, this process is considered independent of the most commonly studied form of autophagy (i.e. macroautophagy). Molecular regulation of crinophagy is unclear, but in beta cells is known to be upregulated by inhibition of insulin secretion. A process different from crinophagy has also been described in beta cells: macroautophagy of insulin granules (but quantitatively minor compared to crinophagy). Since mTORC1/Ulk1 is critical for macroautophagy regulation, do the authors see evidences of macroautophagy in their EM images? (double membrane structures), either of insulin granules or other cytoplasmic components.

3. Please check symbol identification in the figure legends, especially Figures 2E-F. Figure 2 legend describes experiments not shown in the figure.

4. The systemic effects of autophagy inhibitors CQ or 3-MA include changes in glucose tolerance with CQ treatment. Thus, to substantiate the conclusions, the authors should provide evidence that autophagy is inhibited in beta cells. For example, they could show p62 accumulation in beta cells from 3MA-treated mice. This will also strengthen the result shown in Figure 4F, which differs from the results obtained in beta cell lines treated with autophagy inhibitors (DOI: 10.1007/s00125-016-3868-9).

5. Regarding the last comment, the authors provide evidence of a cell-autonomous effect of autophagy inhibition on cell survival (Supp. Figure 5A). This result is key to support some of the manuscript's conclusions. But this results contradicts earlier findings on autophagy and beta cell survival (for example DOI: 10.2337/db12-1474). Please provide an explanation.

6. To follow up on the last comment, the conclusions of the manuscript generally run counter the notion that autophagy is a cytoprotective mechanism in beta cells. While autophagy might be detrimental in bRaKO mice, the loss of autophagy has been consistently shown to be deleterious for beta cells (Atg7 deficiency: Ebato et al., Jung et al, Cell Metab 2008), including experiments with mTORC1 activation, in direct contrast to bRaKO (Bartolome et al., Diabetes 2014). I'm not endorsing one view or the other, but the authors should discuss this point.

7. On the role of S6k on autophagy, the authors rely on a single experiment to confirm "autophagy restoration to normal levels", (Supp. Fig 7A). In this figure LC3-II levels are higher than control,

Moreover, a static measurement of LC3-II is not sufficient to draw conclusions on the status of autophagy. The authors discuss two prior reports in which a S6k/FLIP/Atg3 mechanisms consistent with their conclusions, but there are others that consider S6k as pro-autophagy. Again, this engenders confusion.

8. The second part of the manuscript is clearer, but only indirectly connected with the first, and I am not sure that the two observations belong together. While others have suggested a link between CPE and ER-stress, as discussed in the manuscript, the authors don't think that acute expression of CPE can rescue apoptosis in MIP-bRaKO cells (Fig 5H). However, the authors also report changes in PC1/3; so, perhaps CPE alone is not sufficient to restore insulin processing. Do the authors see an ER-stress phenotype in bRaKO, as reported in other studies that focus on CPE? And if so, have they considered a contribution of ER stress to the cell death phenotype?

Reviewer #2 (Remarks to the Author):

This paper purports to describe the role of mTORC1 in insulin signalling in the pancreatic beta-cell. This is a potentially interesting topic. However, it is impossible to review the paper properly as it is full of errors. Parts of the figures appear to be missing or are incorrectly described. Drug concentrations are not given. The Methods are far too succinct. The paper is riddled with grammatical errors, and ambiguous phrasing. As a consequence, it is often difficult to understand what the authors mean. There are too many examples of the above to list separately. Below I give just a few examples.

A. Examples of grammatical errors, and ambiguous phrasing.

Page 3. 'mTORC1contains exclusively raptor sounds funny'. It sounds as if the mTORC1 complex consists solely of Raptor. Of course it does not – there are many other proteins. So, do they mean that the mTORC1 complex but not MTORC 2 complex contains the Regulatory-associated protein of mTOR (also known as raptor, which is encoded by the RPTOR gene)? Or more broadly that this pathway is the only one that involves Raptor.

Page 12. Line 434. The text states "Mice with targeted deletion of raptor, RIP-Cre... and CaS6K have been previously described". This implies that RIP-Cre, caS6K, etc have all been deleted. I don't think this is what the authors mean.

B. Some comments on the first few pages

The authors need to explain better that RAPTOR associates with eukaryotic initiation factor 4E-binding protein-1 and ribosomal protein S6 kinase (and numerous other proteins). And that its genetic deletion leads to loss of TORC1 activity. Otherwise it is unclear to the reader why genetic deletion of raptor has affects mTORC1.

Page 5. The title of Supplementary Fig 1 titles state it is an analysis of newborn mice. This is not true – A and B are from 30-day old and 120-day old animals respectively.

It is not obvious from Figure 1 that braKO mice exhibited recombination in most beta-cells. This shows insulin glucagon, insulin and GFP. I can only assume that GFP is switched on when Raptor is deleted (by RIP-Cre). What exactly is shown? The mice are referred to as braKO;CAG-GFP - what does this mean? How are these showing beta-cell specific expression?

Line 89. Supplementary Fig 1A,B. does not show phosphorylation, in contrast to what is stated.

C. Some comments on the Methods

Line 431. More details should be given of the human islets.

Line 437. The ages of the mice should be stated. If this varies between figures, the ages should be stated in every figure legend. The light dark cycle under which the mice were maintained should also be stated.

Line 452. The temperature and CO₂/O₂ levels for the culture should be given

Line 454. The temperature at which the insulin secretion experiments were conducted should be given.

Line 458. What was the glucose concentration in these experiments?

Line 456. Define 3-MA

Line 467. What is meant by 'and per 8 weeks'? Is 'per' a typo? Or does it mean the mice were treated every 8 weeks with 3-MA?

Line 468. How was chloroquine administered? Subcutaneous injection? Oral gavage?

Line 469. More details should be given of rapamycin treatment – Were the islets kept in tissue culture? At what glucose concentration? etc.

Line 474. Was there a control for rapamycin injection? i.e. were mice injected with the vehicle alone? This is not stated.

Lines 475-479. This section repeats what is said above in the previous section.

Line 481. What does overnight recovery mean? That the islets were maintained in tissue culture overnight?

Line 494. Strictly speaking the authors have not have measured beta-cell mass as they did not weight the pancreas. They have measured the insulin positive area. This should be corrected.

Line 508. There is little details given here of how the islets were treated.

Line 515. It is not clear what 'equivalent numbers of islets' means.

D. Some comments on Figs. 1 and 2.

Line 742. I am unclear what n=4 per group means. Only 3 replicates are shown. Does each lane represent pooled islets from 4 mice? The same comments apply to other figures (e.g. 2A)

Fig 1B. needs more explanation

Line 744. This should say random blood glucose levels.

Line 745. Fig. 1D. Fed insulin levels. Is this free fed (i.e. random) or were they fasted and then fed at some time point before measurements?

Line 748. Fig. 1G. How was beta-cell mass measured? The Methods says it was measured from the insulin-positive area of the pancreas – so why is it in 'mg' not cm²? The same thing applies to Fig.1K.

Fig 1, H-J. The label is unclear. What is meant by beta-cell positive? In H, is this Ki67 positive cells expressed as percentage of all insulin-positive cells? The same question applies to Fig. 1I and J.

Fig1. J. What does n=4 mean? Were only 4 islets measured? Or was it 4 mice per group. The authors should state both the number of islets/mouse and number of mice.

Fig.1 L, right. The scale bars cannot be 50mM.

Fig.2A. Line 758-60. The cleaved caspase 3 data is very unconvincing. There also seems to be another band (or a piece of dirt) right above it that has been cut off. Which band was measured in the LC-III data. This should be stated in the legend. There is no quantification shown for raptor. The quantification for other proteins is given as a percentage of tubulin. Thus I do not understand the BIMEL/tubulin plot. Given that the BIMEL band is so much more intense than tubulin, why is the BIMEL/tubulin signal less than 1?

Fig. 2C. Some quantification is needed for this to be convincing. i.e the number of lysosomes containing insulin granules needs to be counted.

Fig.2D. Left. This shows actin – not tubulin as claimed. So what are the graphs on the right normalized to – actin or tubulin? Which band is measured for LC-III.

Fig.2E. What was the chlorquine concentration? What is being compared for the p values? What does the hash symbol mean? These should be stated in the legend

Fig.2E. What was the 3-MA concentration? What is being compared for the p values? What does the hash symbol mean? These should be stated in the legend

Fig.1 G shows beta-cell mass – not beta-cell mass, proliferation and Tunel as stated in the legend.

Line 771. There is no Fig.1H (insulin secretion data), despite this being stated in the legend.

Reviewer #3 (Remarks to the Author):

mTOR signaling is an emerging target for maintaining beta cell function and mass in diabetes. However, mechanisms regulating function, survival, and proliferation of beta cells via mTOR signaling are still not clear. In this article, the authors show compelling evidence that mTORC1, a component of mTOR signaling, plays a critical role in maintaining beta cell mass and function. Using multiple genetically engineered mice, the authors dissected out the roles of major components involved in mTORC signaling, including raptor, S6K, Eif4ebp1, and Eif4ebp2 in beta cells. It has been established that reduced beta cell mass is a major pathogenic component of diabetes. Therefore, molecules and signaling pathways regulating beta cell mass are attractive targets for diabetes treatment. The authors' model is supported by well-designed experiments and positive results. The role of mTORC1 in beta cell function, such as insulin processing and beta cell proliferation, has not been examined in vivo before. This is an important contribution to our current efforts on understanding the mechanisms of beta cell dysfunction and reduced beta cell

mass in diabetes. There are a few minor points that should be addressed by the authors to further improve the quality of this important article.

1. Activation of autophagy has been shown to protect beta cells from cell death. However, in raptor knockout mice, activation of autophagy induces beta cell death. The authors should discuss this issue in the discussion section.
2. The role of mTORC1 in glucose-stimulated insulin secretion is still not clear in author's model. The authors should comment on this in the discussion section.

Reviewer #4 (Remarks to the Author):

The manuscript entitled "Genetic dissection of mTORC1 signaling in β -cell" by Blandino-Rosano et al describes the impact of genetic deletion of raptor in beta cells. Using two different raptor ablation models, they find the mice develop insulin-deficient diabetes due to impairment in beta cell mass and function. Increased apoptosis is traced to dysregulation of autophagy. In what could be a very nice set of reconstitution experiments, an attempt is made to trace downstream mTORC1 mediators to determine which pathways are responsible for which outcome. However, the lack of cre-alone controls in the raptor knockout models and the lack of individual-mutants alone controls in the multiple-mutant experiments make interpretation uncertain. It seems likely that the results may be overinterpreted due to the baseline phenotype of these, and the impact of deletion in other tissues. The manuscript is confusingly written and hard to follow. The large amount of work presented may benefit from splitting this into two manuscripts, which would allow a more focused and methodical building of ideas, and better introduction of the many molecules involved, that would improve clarity.

Major concerns:

1. The use of RIPcre and MIPcreERT mice without cre-alone controls greatly weakens the conclusions possible from these models. RIPcre mice express cre in the hypothalamus, and the MIPcreERT mouse has multiple concerns associated with it. These concerns are exaggerated by the fact that in some cases the same parameters are either not measured or had different results. Model-related concerns would be lessened if two different models with different caveats had the same phenotype. The disorganized way the results in these two models are described emphasizes differences more than similarities.
2. bRAKO mice are more insulin sensitive than controls (Figure 3C). Could this be due to hypothalamic deletion? Could some of the beta cell phenotype in this model be due to improved insulin sensitivity?
3. The conclusion on line 124 that the phenotype is due to insulin secretion seems to contradict the findings that beta cell mass, size, proliferation and death were all also impacted
4. The combined genetic model experiments in Figure 3 do not include single mutant results for comparison, and no description is included of the phenotypes of these individual mutants, making it difficult to interpret the results. Which of these are beta cell specific and which are global? Insufficient experimental detail is given.
5. Was islet insulin content reduced in the KO models?
6. Growth hormone increases autophagy in hepatocytes (Zhang et al, PNAS 2015). Please discuss whether GH expression in MIPcreERT mice might have impacted the study results. This is largely mitigated by the interchangeable use of bRAKO mice for these experiments but most experiments are performed in one or the other but not both.
7. Insulin granule morphology looks quite different in the one example shown - please describe whether this was different across multiple beta cells and animals and if so show higher - magnification examples
8. The images in Supplemental Figure 2 do not support the statement that islet architecture was normal at 30 days; in every case the number of non-beta cells appears to be much larger in bRAKO mice than controls. Please quantify the number of islet endocrine cell types for these mice

and discuss possible reasons if different.

9. Images in Supplemental Figure 3 also raise questions. Is Pdx1 non-nuclear in BRAKO mice? Why is MafA staining seen in exocrine cells in both samples? The images are too low magnification and poor pixel density to clearly see the result.

10. Only male mice are used. Please include a reason for this- did female mice not have the same phenotype? Given the sex-dependence of IRS2KO mice this information will be of interest to the field and needs to be included in the manuscript.

11. Since LC3-II is present on autophagosomes, it is not clear to this reviewer why increased LC3II abundance seen with increased survival can lead to a confident conclusion "confirming the contribution of increased autophagic flux to β -cell survival". Please introduce this better. Also, please discuss off-target caveats of NH₄Cl.

12. A diagram summarizing the proposed pathways downstream of Raptor and their effects would be very helpful towards understanding this complicated system.

13. The manuscript is disorganized and difficult to understand. Data are presented haphazardly and not in temporal or logical order. The text at times contradicts itself, in some places saying the effect on mass is exclusively through apoptosis and in other places noting a marked reduction in proliferation. Paragraph structure does not lead to easy understanding of the main results. Consider breaking Results section into more subsections to help organize it. The introduction needs a better description of mTOR downstream effectors to help the reader interpret the data

Minor concerns

1. Bar graph in 1A contains extra 'o's in the y axis. Please clarify meaning of "percentage respect raptor"

2. Please include the age of mice for the immunostaining in panel 1B. The quality of the image on the pdf for review is not high enough to clearly see the result. Consider enlarging the images, improving the clarity or selecting images that are in focus, and including an insert that shows the odd-looking insulin staining in higher magnification and/or in monochrome like the pS6 panel

3. The age of mice used for MIPcreERT deletion cannot be found in the manuscript. Age of BRAKO islets used for insulin secretion assay in Supplemental Figure 4H is not given.

4. The Methods section is disorganized and repetitive. For example, the tamoxifen section is included twice, and there is no description of which mice received tamoxifen injections.

5. Figure font size too small to read even when blown up to max size on my monitor

6. If controls are repeated in Figure 2E and F this should be stated in the legend

7. Supplemental Figure 5D concentration of glucose used should be included in the legend rather than just 'low' and 'high'

8. Typographical errors: line 87: form, fig 2A 'clev', line 120 'increase',

Reviewers' comments:

Reviewer #1 (Remarks to the Author):

The authors explore the effects of raptor knockout in beta cells (bRaKO). They describe a phenotype of hypoinsulinemia and hyperglycemia, associated with reduced beta cell mass. The authors suggest that autophagy has deleterious effects in for beta cell survival in bRaKO. They also analyze the contribution of mTORC1 targets, uncovering the specific role of 4ebp2 for CPE translation and insulin processing. The data are mostly clear, but I am confused by the conclusions: specifically, the authors' conclusions on autophagy are in contrast to existing literature and other models of mTOR gain-of-function. The second part seems to be a bit of an afterthought, and is not well integrated with the first. The work is well done, but the conclusions a bit fuzzy.

We are delighted to hear for the positive comments and appreciation of our work. We have made a major effort to address these concerns and provide a better integration of the second part.

1. Figure 2B shows percentage of LC3-positive beta cells (Now Fig. 2C) (in Lc3-gfp transgenic mice). LC3 should be expressed in all cells, but it appears to be detected only detected in few cells. Is there an explanation for this? LC3 expression alone is not sufficient to conclude that there is autophagy. It would be useful to obtain higher magnification images to monitor subcellular distribution of LC3 (diffuse in cytoplasm vs. accumulation in puncta) which is characteristic of LC3-I and LC3-II respectively (the latter is representative of ongoing autophagy, even though a single measurement doesn't provide information on rates of autophagy).

As suggested by the reviewer, we included a higher magnification images to assess the cellular distribution of LC3. These images show that dispersed β -cells from control mice exhibit low cytoplasmic GFP fluorescence (read out of LC3 expression) (New Fig. 2C). In contrast, cytoplasmic GFP fluorescence was elevated in the majority of *braKO* β -cells. A fraction of *braKO* β -cells showed distinct cytoplasmic puncta consistent with autophagosome formation. We now include new data showing that the fraction of β -cells with LC3-GFP puncta formation was augmented in *braKO* mice at 40 days (Fig. 2C). We agree with the reviewer that we cannot assess rates of autophagy and therefore have removed statements using this term.

2. Figure 2C (Now Fig. 2A). The authors show increased numbers of insulin granule-containing lysosomes (crinophagy). Generally, this process is considered independent of the most commonly studied form of autophagy (i.e. macroautophagy). Molecular regulation of crinophagy is unclear, but in beta cells

is known to be upregulated by inhibition of insulin secretion. A process different from crinophagy has also been described in beta cells: macroautophagy of insulin granules (but quantitatively minor compared to crinophagy). Since mTORC1/ULK1 is critical for macroautophagy regulation, do the authors see evidences of macroautophagy in their EM images? (double membrane structures), either of insulin granules or other cytoplasmic components.

The reviewer raises a very important point. The contribution of macroautophagy versus microautophagy in β -cells is still unclear. There are several pieces of evidence suggesting that *raptor* inactivation in β -cells induces increased macroautophagy: 1. Decreased ULK phosphorylation; 2. Increased puncta formation and increased LC3-II by immunoblotting; and 3. Data showing p62 accumulation after NH_4Cl , a new staining showing p62 accumulation after 3-MA and CQ treated mice *in vivo* and a partial rescue of β -cell mass by these agents. However, we failed to see evidence of macroautophagy (double membrane structures) by EM at 30 and 60 days. We do not have a clear explanation for the lack of detection of macroautophagy by EM but there are several points that could contribute to this finding: 1. Formation of autophagosomes is rarely detected under steady-state conditions in β -cells as previously discussed^{1, 2}, and 2. There are major differences in autophagic vacuole formation between β -cells and other organs. In particular, starvation-induced autophagy is present in β -cells but perhaps is less prominent than in other tissues. While overnight starvation increases autophagic vacuole in other organs such as liver and muscle³, β -cells show no prominent increase in autophagic vacuoles after starvation². This response is similar to the lack of increased autophagy in neuronal tissues under starvation conditions. Taken together, the current evidence support the concept that while starvation-induced autophagy is present in β -cells, it is less prominent than in other tissues. Therefore, it is possible that macroautophagy detection by EM is more elusive in *raptor deletion* as this model mimics conditions of nutrient starvation. It is also possible that the ratio between macro and microautophagy is modulated differently in the *β raKO* mice, a model of chronic “genetic starvation” in an environment with normal nutrients. A caveat in the Discussion includes this explanation.

3. Please check symbol identification in the figure legends, especially Figures 2E-F (Now Fig. 3D-E). Figure 2 legend describes experiments not shown in the figure.

This is now corrected. Thank you.

4. The systemic effects of autophagy inhibitors CQ or 3-MA include changes in glucose tolerance with CQ treatment. Thus, to substantiate the conclusions, the authors should provide evidence that autophagy is inhibited in beta cells. For example, they could show p62 accumulation in beta cells from 3MA-treated mice.

This will also strengthen the result shown in Figure 4F (Now Fig. 3F), which differs from the results obtained in beta cell lines treated with autophagy inhibitors (DOI: 10.1007/s00125-016-3868-9).

We appreciate that the reviewer suggested this experiment. Indeed, these experiments showed that there was accumulation of p62 in islets from 3MA and CQ treated groups providing evidence that autophagy was inhibited by the treatment (New Fig. 3C). We thank the reviewer for suggesting this important experiment as we believe that these data now strengthen our conclusions by providing evidence that autophagy was inhibited by these agents. Thank you.

5. Regarding the last comment, the authors provide evidence of a cell-autonomous effect of autophagy inhibition on cell survival (Supp. Figure 5A). This result is key to support some of the manuscript's conclusions. But this results contradicts earlier findings on autophagy and beta cell survival (for example DOI: 10.2337/db12-1474). Please provide an explanation.

The reviewer brings an important point. The manuscripts referred by the reviewer showed that treatment of female diabetic *Akita* mice with rapamycin improved diabetes, increased pancreatic insulin content and prevented β -cells apoptosis. This manuscript further showed that this was associated to increased autophagy and conclude that the beneficial effects of rapamycin in this context strictly depend on autophagy. There are several differences that could explain the contradictory findings between the rapamycin treated *Akita* mice and our *braKO* models: 1. Rapamycin treatment is not equivalent to *raptor* deletion. It is known that long-term rapamycin treatment also inhibit mTORC2 and therefore some of these results could be explained by concomitant inhibition of mTORC1 and mTORC2. 2. Rapamycin has been shown to inhibit primarily the effect on S6K and less 4E-BPs^{4,5}. 3. More importantly, *Akita* mice exhibit apoptosis induced by ER stress due to misfolding of mutant insulin and therefore conditions that change insulin biosynthesis could alter these responses. To this end, we have shown that insulin content is reduced in *braKO* β -cells (Fig. 5D, E and F) and it is possible that the beneficial effects of rapamycin could be caused in part by decreasing insulin biosynthesis.

6. To follow up on the last comment, the conclusions of the manuscript generally run counter the notion that autophagy is a cytoprotective mechanism in beta cells. While autophagy might be detrimental in bRaKO mice, the loss of autophagy has been consistently shown to be deleterious for beta cells (Atg7 deficiency: Ebato et al., Jung et al, Cell Metab 2008), including experiments with mTORC1 activation, in direct contrast to bRaKO (Bartolome et al., Diabetes 2014). I'm not endorsing one view or the other, but the authors should discuss this point.

We are aware of these manuscripts and agree with the reviewer that this point needs more clarification. These manuscripts suggest that basal autophagy has a cytoprotective mechanism in β -cells. In contrast to these models, *β raKO* mice exhibit a detrimental effect from extreme and chronic enhanced autophagy in β -cells. Taken together, these studies are consistent with the concept that fine-tuning of autophagy is essential for proper β -cell homeostasis and conditions of extreme upregulation or inhibition of autophagy can negatively impact β -cells. To clarify this concern we added a new paragraph in the Discussion. Thank for your comments.

7. On the role of S6k on autophagy, the authors rely on a single experiment to confirm “autophagy restoration to normal levels”, (Supp. Fig 7A) (Now Supplemental Fig. 9A). In this figure LC3-II levels are higher than control, Moreover, a static measurement of LC3-II is not sufficient to draw conclusions on the status of autophagy. The authors discuss two prior reports in which a S6k/FLIP/Atg3 mechanisms s consistent with their conclusions, but there are others that consider S6k as pro-autophagy. Again, this engenders confusion.

The reviewer brings an important point. To answer this concern, it is important to acknowledge that the role of S6K in autophagy is still not well understood ⁶. There is evidence showing both activation and inhibition of autophagy by S6K. Therefore, we attempted to provide some potential explanations for these discrepancies. A caveat in the Discussion now reads:

“These results are in marked contrast with published data showing that phosphorylation of S6K is considered as pro-autophagy ^{7, 8, 9}. The effect of S6K in autophagy is controversial ⁶ but discrepancies in the literature could be explained in part by: 1. Different mechanisms and duration of mTOR inhibition (amino acid deprivation/rapamycin vs genetic inactivation and chronic vs. acute). 2. Our S6K overactivation model provides a non-physiological state of permanent S6K activation. 3. Previous work showing that S6K promotes autophagy was obtained in a model of genetic deletion of mTOR in flies ⁹. However, in this model mTOR inactivation disrupts both mTORC1 and mTORC2 and mTORC2/Akt-FoxO pathway may be also involved in the control of autophagy ¹⁰. More studies should be designed to clarify the role of S6K in autophagy in β -cells and other tissues.”

8. The second part of the manuscript is clearer, but only indirectly connected with the first, and I am not sure that the two observations belong together. While others have suggested a link between CPE and ER-stress, as discussed in the manuscript, the authors don't think that acute expression of CPE can rescue apoptosis in MIP-bRaKO cells (Fig 5H) (Now Supplemental Fig. 13E). However, the authors also report changes in PC1/3; so, perhaps CPE alone is not sufficient to restore insulin processing.

The reviewer raises the possibility that CPE alone is not sufficient to restore insulin processing. Normalization of CPE and not PC1/3 in *βraKO;Eif4ebp2^{-/-}* (Fig. 5G and Supplemental Fig. 11A) indicate that CPE alone is sufficient to restore insulin processing but not survival in *βraKO* mice suggesting a disconnection between these processes. We include a caveat in the Discussion.

Do the authors see an ER-stress phenotype in bRaKO, as reported in other studies that focus on CPE? And if so, have they considered a contribution of ER stress to the cell death phenotype?

We agree that ER stress could be part of the phenotype. Therefore, we assessed CHOP levels by western blot in islets from 30 day-old *βraKO* and control mice (new Supplemental Figure 13A). CHOP levels were not different between control and *βraKO* islets. We also assessed CHOP expression at a single cell level by flow cytometry. CHOP levels (MFI) in insulin positive cells in control and *βraKO* cells were comparable (new Supplemental Fig. 13B).

Reviewer #2 (Remarks to the Author):

This paper purports to describe the role of mTORC1 in insulin signalling in the pancreatic beta-cell. This is a potentially interesting topic. However, it is impossible to review the paper properly as it is full of errors. Parts of the Figures appear to be missing or are incorrectly described. Drug concentrations are not given. The Methods are far too succinct. The paper is riddled with grammatical errors, and ambiguous phrasing. As a consequence, it is often difficult to understand what the authors mean. There are too many examples of the above to list separately. Below I give just a few examples.

Thank you for the comments. We addressed the Figure concerns, added drug concentrations and improved the Methods and ambiguous phrasing in the manuscript.

A. Examples of grammatical errors, and ambiguous phrasing.

Page 3. 'mTORC1contains exclusively raptor sounds funny'. It sounds as if the mTORC1 complex consists solely of Raptor. Of course it does not – there are many other proteins. So, do they mean that the mTORC1 complex but not mTORC 2 complex contains the Regulatory-associated protein of mTOR (also known as raptor, which is encoded by the RPTOR gene)? Or more broadly that this pathway is the only one that involves Raptor.

This sentence has been corrected and we now describe the different components of the mTORC1 complex in the Introduction.

Page 12. Line 434. The text states "Mice with targeted deletion of raptor, RIP-Cre... and CaS6K have been previously described". This implies that RIP-Cre, caS6K, etc have all been deleted. I don't think this is what the authors mean.

This paragraph has been corrected

B. Some comments on the first few pages

The authors need to explain better that RAPTOR associates with eukaryotic initiation factor 4E-binding protein-1 and ribosomal protein S6 kinase (and numerous other proteins). And that its genetic deletion leads to loss of TORC1 activity. Otherwise it is unclear to the reader why genetic deletion of raptor has affects mTORC1.

This is now better explained in the Introduction.

Page 5. The title of Supplementary Fig 1 titles state it is an analysis of newborn

mice. This is not true – A and B are from 30-day old and 120-day old animals respectively.

Supplemental Fig. 1 has been replaced after restructuring the manuscript. The data in the original Supplemental Fig. 1 is now included in other Figures.

It is not obvious from Figure 1 that β raKO mice exhibited recombination in most beta-cells. This shows insulin glucagon, insulin and GFP. I can only assume that GFP is switched on when Raptor is deleted (by RIP-Cre). What exactly is shown? The mice are referred to as β raKO;CAG-GFP - what does this mean? How are these showing beta-cell specific expression?

We apologize for not explaining this better. In these studies we used a reporter mouse model, B6.Cg-Gt(ROSA)26Sor^{tm6(CAG-ZsGreen1)Hze/J} (CAG-GFP in the manuscript) in which EGFP expression is regulated by a generalized promoter after Cre-mediated recombination. We generated β raKO mice in the CAG-GFP background to examine the recombination efficiency of the RIP-Cre used to inactivate raptor (β raKO; CAG-GFP). Therefore, cells expressing RIP-Cre are expected express EGFP and lack Raptor. Supplemental Fig. 1B shows: 1. 95% of the cells are positive for insulin and GFP suggesting that Cre-mediated recombination occurred in the majority of the β -cells. Deletion of raptor in the majority of β -cells was also confirmed by immunoblotting (Fig. 1A). 2. We did not see any GFP staining in α -cells or acinar tissue confirming that the insulin cells are not dedifferentiating.

Line 89. Supplementary Fig 1A,B. does not show phosphorylation, in contrast to what is stated.

This has now been corrected.

C. Some comments on the Methods

Line 431. More details should be given of the human islets.

Supplemental Table 1 now contains information of the human islets donors.

Line 437. The ages of the mice should be stated. If this varies between Figures, the ages should be stated in every figure legend.

Age of the mice are now included in all the Figure legends.

The light dark cycle under which the mice were maintained should also be stated.

All animals were maintained on a 12h light-dark cycle. This is now included in the

Methods.

Line 452. The temperature and CO₂/O₂ levels for the culture should be given

Islets were maintained at 37°C in an atmosphere of 20% oxygen and 5% CO₂. This is now included in the Methods.

Line 454. The temperature at which the insulin secretion experiments were conducted should be given.

This is now included in the Methods.

Line 458. What was the glucose concentration in these experiments?

This is now included.

Line 456. Define 3-MA

Now added to the Results and Methods sections.

Line 467. What is meant by 'and per 8 weeks'? Is 'per' a typo? Or does it mean the mice were treated every 8 weeks with 3-MA?

This typo is now corrected.

Line 468. How was chloroquine administered? Subcutaneous injection? Oral gavage?

Chloroquine was injected intraperitoneally. This is now included.

Line 469. More details should be given of rapamycin treatment – Were the islets kept in tissue culture? At what glucose concentration? etc.

This has been now included:

“For rapamycin (LC Laboratories) treatment *in vivo*, rapamycin was dissolved in 100% ethanol and stored at –20°C. Stock solution was further diluted in an aqueous solution of 5.2% Tween 80 and 5.2% PEG 400 (final ethanol concentration, 2%). Control and *Eif4ebp2*^{-/-} mice were injected with rapamycin intraperitoneally (2 mg/kg body weight) or vehicle (2% ethanol) every other day for 9 days. Islets from these mice were isolated and immediately lysed after isolation.”

“For rapamycin treatment *in vitro*, islets were cultured in RPMI containing 5 mM glucose and rapamycin (Sigma) (30 nM) for 24 or 48 hours as indicated in the

corresponding Figure Legends.”

Line 474. Was there a control for rapamycin injection? i.e. were mice injected with the vehicle alone? This is not stated.

Yes, control mice were injected with vehicle (2% ethanol). This is now included in the Methods: paragraph entitled “Preparation and *in vivo* treatment with different agents”.

Lines 475-479. This section repeats what is said above in the previous section.

This section has been removed. Thanks.

Line 481. What does overnight recovery mean? That the islets were maintained in tissue culture overnight?

We failed to clearly explain this. Overnight recovery means that islets were cultured overnight in 5 mM glucose and 10% FBS after the isolation. This is now included.

Line 494. Strictly speaking the authors have not have measured beta-cell mas as they did not weight the pancreas. They have measured the insulin positive area. This should be corrected.

All β -cell mass data was obtained by multiplying the insulin/acinar ratio by the pancreas weight except for neonates.

Line 508. There is little details given here of how the islets were treated.

More details have been now added.

Line 515. It is not clear what ‘equivalent numbers of islets’ means.

Corrected.

D. Some comments on Figs. 1 and 2.

Line 742. I am unclear what $n=4$ per group means. Only 3 replicates are shown. Does each lane represent pooled islets from 4 mice? The same comments apply to other Figures (e.g. 2A)

We apologize for the confusion. Every lane represents data from one mouse and is considered as one biological replicate. Therefore, $n=4$ means that the analysis was done in quadruplicate or included data from four mice per condition. This is

now stated in the Figure legends “A representative image from four independent experiments is included and each lane shows the expression levels from one mouse.”

Fig 1B. needs more explanation

This paragraph has been rewritten.

Line 744. This should say random blood glucose levels.

Fixed

Line 745. Fig. 1D (Now Fig. 1C). Fed insulin levels. Is this free fed (i.e. random) or were they fasted and then fed at some time point before measurements?

Fed insulin is expressed now as Random fed serum insulin in the Figure legend.

Line 748. Fig. 1G (Now Fig. 1D). How was beta-cell mas measured? The Methods says it was measured from the insulin-positive area of the pancreas – so why is it in ‘mg’ not cm²? The same thing applies to Fig.1K (Now Fig. 1H).

It was multiplied by the pancreas weight and expressed in mg except for neonates.

Fig 1, H-J (Now Fig. 1E-G). The label is unclear. What is meant by beta-cell positive? In H, is this Ki67 positive cells expressed as percentage of all insulin-positive cells? The same question applies to Fig. 1I and J.

The reviewer is correct; we express the data as percentage of insulin positive cells. We now replaced β -cell positive by insulin positive.

Fig1. J. What does n=4 mean? Were only 4 islets measured? Or was it 4 mice per group. The authors should state both the number of islets/mouse and number of mice.

n=4 means islets were isolated from four mice. Islets from each mouse were cultured independently for 24 hours in 5 mM or 24 mM glucose.

Fig.1 L, right. The scale bars cannot be 50mM.

Corrected to μ M. Thank you.

Fig.2A (now Supplemental Fig. 6C). Line 758-60. The cleaved capsase 3 data is very unconvincing. There also seems to be another band (or a piece of dirt) right

above it that has been cut off. Which band was measured in the LC-III data. This should be stated in the legend. There is no quantification shown for raptor. The quantification for other proteins is give as a percentage of tubulin. Thus I do not understand the BIMEL/tubulin plot. Given that the BIMEL band is so much more intense than tubulin, why is the BIMEL/tubulin signal less than 1?

Cleaved Caspase 3 always has a weak signal in islet samples. We now include a better image for cleaved caspase 3. In addition, increased cleaved Caspase 3 by immunoblotting was also confirmed by assessment of active Caspase 3 by flow cytometry (Supplemental Fig. 13E). We hope that the reviewer finds these data now convincing.

LC3 exists in two forms; a cytosolic form of LC3 conjugated to phosphatidylethanolamine (LC3-I) and a form that is recruited to autophagosomal membranes (LC3-II). To assess autophagy, total amount of LC3-II should be evaluated and compared to a loading control. LC3-II (lower band of the two LC3 bands) was adjusted to tubulin or actin as loading controls and expressed as LC3-II/tubulin/actin. We have included labels for the LC3-I and LC3-II bands in Fig. 3A and Supplemental Fig. 6C. These changes are also described in Figure Legends.

Quantification for raptor is now added to new Supplemental Fig. 6C.

BIM_{EL} quantification is the average of 4 independent lanes corresponding to 4 different mice (lysate from one mouse in each lane). The original image has been replaced by a more representative image and quantification BIM_{EL} showed no difference.

Fig. 2C (now Fig. 2A). Some quantification is needed for this to be convincing. i.e the number of lysosomes containing insulin granules needs to be counted.

Thanks for this comment. We have performed the analysis as requested by the reviewer and these data is now included in a new Fig. 2A. In addition, we include more images of these structures at different magnifications (New Fig. 2B) as suggested by another reviewer.

Fig.2D (now Fig. 3A). Left. This shows actin – not tubulin as claimed. So what are the graphs on the right normalized to – actin or tubulin? Which band is measured for LC-III.

Thanks for this comment. It should be Actin instead. We measured LC3-II (lower band) as described above. This information is now in Fig. 3A.

Fig.2E (now Fig. 3D). What was the chlorquine concentration? What is being

compared for the p values? What does the hash symbol mean? These should be stated in the legend

Fig.2F (now Fig. 3E). What was the 3-MA concentration? What is being compared for the p values? What does the hash symbol mean? These should be stated in the legend.

Concentrations are now included in the Figure Legend. *P* value symbols are now clearly defined in the Figure Legends. Thank you.

Fig.2G shows beta-cell mass – not beta-cell mass, proliferation and Tunel as stated in the legend.

This has been now corrected.

Line 771. There is no Fig.1H (insulin secretion data), despite this being stated in the legend.

Thank you. This has been now corrected

Reviewer #3 (Remarks to the Author):

mTOR signaling is an emerging target for maintaining beta cell function and mass in diabetes. However, mechanisms regulating function, survival, and proliferation of beta cells via mTOR signaling are still not clear. In this article, the authors show compelling evidence that mTORC1, a component of mTOR signaling, plays a critical role in maintaining beta cell mass and function. Using multiple genetically engineered mice, the authors dissected out the roles of major components involved in mTORC signaling, including raptor, S6K, Eif4ebp1, and Eif4ebp2 in beta cells. It has been established that reduced beta cell mass is a major pathogenic component of diabetes. Therefore, molecules and signaling pathways regulating beta cell mass are attractive targets for diabetes treatment. The authors' model is supported by well-designed experiments and positive results. The role of mTORC1 in beta cell function, such as insulin processing and beta cell proliferation, has not been examined in vivo before. This is an important contribution to our current efforts on understanding the mechanisms of beta cell dysfunction and reduced beta cell mass in diabetes. There are a few minor points that should be addressed by the authors to further improve the quality of this important article.

We really appreciate that the reviewer considers our experiments well-designed, and that our article makes an important contribution. We agree that there is no other study with these amounts of genetically engineered different mice and data in mTORC1 in β -cell or other tissues.

1. Activation of autophagy has been shown to protect beta cells from cell death. However, in raptor knockout mice, activation of autophagy induces beta cell death. The authors should discuss this issue in the discussion section.

This point was raised from Reviewer 1 (Please see answers to comments 1, 2, 5, 6 and 7 from Reviewer #1).

2. The role of mTORC1 in glucose-stimulated insulin secretion is still not clear in author's model. The authors should comment on this in the discussion section.

We agree with the reviewer in that, the role of mTORC1 in glucose-stimulated insulin secretion is still not clear. We believe that this is a very interesting phenotype that we would like to pursue and plan to design experiments to further explore this. We had begun to unravel these mechanisms by assessing intracellular calcium levels. However, the complexity of discovering the mechanisms for the defects in intracellular calcium observed in this model would probably require extensive work and would likely involve at least a year of work. In addition, the eight figures and fifteen Supplemental Figures in the current manuscript leave no space for more information. We hope that the reviewer

understands these limitations and added some preliminary results on this subject to Discussion:

“The lack of changes in β -cell mass in *MIP- β raKO^{ff}* mice allowed us to identify an important role of mTORC1 on insulin secretion and preliminary studies showed that *MIP- β raKO^{ff}* mice exhibit reduced intracellular calcium after glucose stimulation (unpublished data). This suggests that events proximal to calcium influx are involved but it is possible that other steps contribute to this phenotype.”

Reviewer #4 (Remarks to the Author):

The manuscript entitled “Genetic dissection of mTORC1 signaling in β -cell” by Blandino-Rosano et al describes the impact of genetic deletion of raptor in beta cells. Using two different raptor ablation models, they find the mice develop insulin-deficient diabetes due to impairment in beta cell mass and function. Increased apoptosis is traced to dysregulation of autophagy. In what could be a very nice set of reconstitution experiments, an attempt is made to trace downstream mTORC1 mediators to determine which pathways are responsible for which outcome. However, the lack of cre-alone controls in the raptor knockout models and the lack of individual-mutants alone controls in the multiple-mutant experiments make interpretation uncertain. It seems likely that the results may be overinterpreted due to the baseline phenotype of these, and the impact of deletion in other tissues. The manuscript is confusingly written and hard to follow. The large amount of work presented may benefit from splitting this into two manuscripts, which would allow a more focused and methodical building of ideas, and better introduction of the many molecules involved, that would improve clarity.

Thank you for the comments. We really appreciate that the reviewer considers the reconstitution experiments a very nice set of studies. We have addressed all the concerns including the lack of Cre-alone controls and lack of individual mutants. In addition we have made a significant attempt to improve the manuscript. We also agree in that there is a large amount of work presented and at some point considered to split the manuscript in two. However, we believe that this would decrease the impact of the manuscript. In addition, we have followed the recommendation of the Associate Editor to keep the manuscript in one and work on the structure of the manuscripts to improve the readability of the paper.

Major concerns:

1. The use of RIPcre and MIPcreERT mice without cre-alone controls greatly weakens the conclusions possible from these models. RIPcre mice express cre in the hypothalamus, and the MIPcreERT mouse has multiple concerns associated with it.

The reviewer brings a very important point. This is a limitation for all the current studies using Cre-lox technology with available Cre-lines. At the time we started the experiments (2010), we were unaware of some of the limitations of Cre lines used in this manuscript. In response to the reviewer comments, we have emphasized some of the results and performed the following studies to address this concern: 1. We performed studies in the β raptor^{f/+} (heterozygous inactivation of raptor in β -cells) mice. These studies showed that glucose tolerance and insulin sensitivity is normal in β raptor^{f/+} (new Supplemental Figure 14B and C). 2. Figure 1B shows that in contrast to β raKO, random glucose in β raptor^{f/+} is

identical to that of controls. 3. In addition, we performed experiments to control for the *MIP-CreERT* transgene. These studies included *MIP-Cre* β *raKO*^{ff} injected with corn oil as control. The results show that glucose tolerance in *MIP-Cre* β *raKO*^{ff} injected with corn oil is comparable to that of the controls (*raptor*^{ff}) but different than that of *MIP-Cre* β *raKO*^{ff} injected with tamoxifen (new Supplemental Fig. 1L-M and new Supplemental Fig. 6B). In addition to the experimental data we believe that consistency of the results in two different Cre models (*RIP-Cre* and *MIP-CreERT*) supports the conclusions of the study. Interestingly the *RIP-Cre* (Tg(Ins2-cre)^{Herr}) used in these studies does not contain the hGH minigene and therefore imply that the results obtained are unlikely explained by excess of hGH. Finally, the *RIP-Cre* transgene was included in β *raKO*, β *raKO*;*caS6K*, β *raKO*;*Eif4ebp2*^{-/-} and β *raKO*;*caS6K*;*Eif4ebp2*^{-/-} suggesting that the differences between these groups are not likely explained by the presence of *RIP-Cre* transgene. We hope that the reviewer understands that we made a significant attempt to address the concerns experimentally and that this information supports the interpretation of the data. These issues could be resolved with the new knock-in Cre-lines but it would take a significant amount of time to repeat some of the experiments.

These concerns are exaggerated by the fact that in some cases the same parameters are either not measured or had different results. Model-related concerns would be lessened if two different models with different caveats had the same phenotype. The disorganized way the results in these two models are described emphasizes differences more than similarities.

We apologize for not being sufficiently clear in strengthening the similarities of the two-mTORC1 deficient mouse models. In fact, the *MIP- β raKO*^{ff} model confirms the results obtained using the β *raKO* model. Based on the reviewer's concern, we decided to structure Fig. 1 to show the phenotype of both models making it much easier for the reader to compare them.

Fig. 1 now clearly shows that both models recapitulate the major phenotypes:

- 1) Fed hyperglycemia
- 2) Fed hypoinsulinemia
- 3) Apoptosis measured by TUNEL
- 4) Glucose-stimulated insulin secretion

In addition, there are more phenotypic similarities throughout the manuscript including:

- 1) Increased autophagy
- 2) Increased proinsulin content

The only differences found between the models are that the *MIP- β raKO*^{ff} mice have no alteration in proliferation and β -cell mass while the β *raKO* mice exhibit a dramatic reduction in proliferation and β -cell mass. We believe this is not

unexpected as the *βraKO* mice lack Raptor during embryonic stages and at early postnatal development when maturation occurs. On the other hand, inactivation of Raptor *MIP-βraKO^{ff}* occurs in mature β-cells. One could argue that the severity of the *MIP-βraKO^{ff}* phenotype is not as remarkable as in the *βraKO* model, although the successful inactivation of raptor was comparable. However, we only followed these mice for 8 weeks post-TMX until alterations in glucose homeostasis were present. It is possible that longer follow-up would show a more severe phenotype. A caveat mentioning this has been included in the Discussion.

2. bRAKO mice are more insulin sensitive than controls (Figure 4C). Could this be due to hypothalamic deletion? Could some of the beta cell phenotype in this model be due to improved insulin sensitivity?

The reviewer raises an important point. The purpose of assessing insulin sensitivity in these mice was to evaluate the possibility that changes in glucose homeostasis were caused by insulin resistance. Interestingly, we found that *βraKO* mice were rather more insulin sensitivity. While there is a possibility that insulin sensitivity could be due to hypothalamic expression, we believe that the β-cell phenotype is not a consequence of changes in insulin sensitivity in *βraKO* mice. The increase in insulin sensitivity would ameliorate the β-cell phenotype, which is already severe. In addition, the inclusion of the *RIP-Cre* in *βraKO*, *βraKO;caS6K* *βraKO;Eif4ebp2^{-/-}* and *βraKO;caS6K;Eif4ebp2^{-/-}* groups suggest that the differences in the β-cell phenotype between these groups is not due to improved insulin sensitivity. Finally, lack of alterations in weight in both the *RIP* or *MIP* models suggests that it is unlikely that hypothalamic expression is a major factor in the *βraKO* phenotype.

3. The conclusion on line 124 that the phenotype is due to insulin secretion seems to contradict the findings that beta cell mass, size, proliferation and death were all also impacted

We agree with the reviewer. This sentence has been modified.

4. The combined genetic model experiments in Figure 3 do not include single mutant results for comparison, and no description is included of the phenotypes of these individual mutants, making it difficult to interpret the results. Which of these are beta cell specific and which are global? Insufficient experimental detail is given.

The reviewer is correct in that we failed to include data on the single mutants. Phenotypes of individual mutants for *cas6K*, *Eif4ebp1^{-/-}* and *Eif4ebp2^{-/-}* have been already published^{11, 12}. *caS6K* is β-cells specific transgenic mouse and *Eif4ebps* mutants are total body knock out. We now include the description of the individual mutants to help with the interpretation of the results. To avoid any

confusion about β -cells specific vs. global, we now include more information in the Methods and Results section.

5. Was islet insulin content reduced in the KO models?

The reviewer brings an important question. In response to the reviewer's concern, we assessed insulin content in single β -cells using flow cytometry (New Fig. 5E). We believe that this is the best approach to measure insulin content, as the loss of β -cells makes difficult to assess insulin content per islet. New Figure 5E shows that at 30 days, the insulin content was already reduced in *braKO* cells confirming data shown in Fig. 5D by staining and 5F by immunoblotting. The extent to which this is modified by genetic reconstitution is an interesting question but was not explored. We believe this is beyond the scope of this manuscript, which is already very dense. I hope the reviewer understand that these studies will add another layer of complexity to the manuscript.

6. Growth hormone increases autophagy in hepatocytes (Zhang et al, PNAS 2015). Please discuss whether GH expression in MIPcreERT mice might have impacted the study results. This is largely mitigated by the interchangeable use of bRAKO mice for these experiments but most experiments are performed in one or the other but not both.

The reviewer brings an important point related to the hGH minigene used for some of the Cre lines. In contrast to the *MIP-CreERT* model, the *RIP-Cre* from Herrera does not contain the hGH minigene. Given that the autophagy phenotype is present in both models, we believe that GH expression is unlikely to play a role. We now provide data from *MIP-braKO^{ff}* injected with corn oil as controls showing no metabolic abnormalities indicating that GH expression have not impacted the glucose homeostasis results (Fig. 1L, M and new Supplemental Fig. 6B). We agree with the reviewer that the use of both models mitigates the hGH concerns and provide an explanation of the differences of the two models above (See Answer to comment 1). We cannot exclude the possibility that GH could ameliorate the loss of β -cell mass defect in *MIP-braKO^{ff}* model. We now include this caveat in the Discussion. Thank you for this comment.

7. Insulin granule morphology looks quite different in the one example shown-please describe whether this was different across multiple beta cells and animals and if so show higher-magnification examples

Thanks for the comment. We did not find differences in insulin granule morphology across multiple β -cells and animals (30 and 60 days of age). We now include more images of insulin granules from other cells and provide a higher-magnification as requested by the reviewer (new Fig. 2B).

8. The images in Supplemental Figure 2 do not support the statement that islet architecture was normal at 30 days; in every case the number of non-beta cells appears to be much larger in *bRAKO* mice than controls. Please quantify the number of islet endocrine cell types for these mice and discuss possible reasons if different.

We agree with the reviewer in that some Figures show an apparent increase in non- β endocrine cells and now we add this caveat to the manuscript. “Normal islet architecture” was used to describe that islets contain a β -cell core surrounded by non- β -cells. Quantification of islet endocrine cells is difficult to achieve by morphometry. To do this, non- β endocrine cells will have to be quantified in serial sections throughout the pancreas. However, we addressed the issue of increase in endocrine cells by indirectly assessing α -cells (majority of non- β endocrine cells) using total pancreatic glucagon content. This experiment shows that glucagon content in *braKO* mice is comparable to that of controls, indirectly suggesting that there is a similar mass of α -cells (data not shown in the manuscript). These data is included for the reviewer (see below) but we will be happy to add this Figure in the manuscript if the reviewer considers necessary. Finally, while there is a possibility for an increase in non- β endocrine cells in *braKO* mice, the similarities of phenotypes between *braKO* and *MIP-braKO*^{ff} models suggest that this is unlikely to a role in the major findings of the paper.

Figure 1. Pancreatic glucagon content in 30 day-old control and *braKO* mice. ($n=4$, $P=n.s.$).

9. Images in Supplemental Figure 3 (Now Supplemental Fig. 4) also raise questions. Is *Pdx1* non-nuclear in *BRAKO* mice? Why is *MafA* staining seen in exocrine cells in both samples? The images are too low magnification and poor pixel density to clearly see the result.

Thank you for pointing that out. We now include new images at higher magnification and better resolution. PDX1 is mainly nuclear and *MafA* staining is only shown in β -cells. Acinar staining in the previous images was due to an increased and saturated background of the red channel.

10. Only male mice are used. Please include a reason for this- did female mice not have the same phenotype? Given the sex-dependence of *IRS2KO* mice this

information will be of interest to the field and needs to be included in the manuscript.

We now include data replicating the hyperglycemic phenotype in Females (New Supplemental Fig. 14A). We have also validated some of the changes in autophagy and insulin processing (data not shown).

11. Since LC3-II is present on autophagosomes, it is not clear to this reviewer why increased LC3II abundance seen with increased survival can lead to a confident conclusion “confirming the contribution of increased autophagic flux to β -cell survival”. Please introduce this better. Also, please discuss off-target caveats of NH₄CL.

We apologize for not clearly introducing these experiments. Accumulation of autophagy markers such as p62 or LC3 after the treatment with autophagy blockers is an established endpoint to measure autophagy flux¹³. The increase in LC3II and p62 post-treatment with autophagy blockers in our model is consistent with the concept of increased autophagic flux in the mutant β -cells. Concomitant decrease in TUNEL after treatment with autophagy inhibitors also provides evidence for the contribution of autophagy to β -cell survival. We have now introduced this better.

NH₄Cl blocks autophagy flux by inducing changes in lysosomal pH. General inhibitors of lysosomal proteases (e.g., bafilomycin A1, NH₄Cl, leupeptin) also block the degradation of proteins delivered to lysosomes by other autophagic and endosomal pathways¹³. Potential off target effects are increase in cell protein content and cell size¹⁴. The validation of the results with chloroquine and 3MA suggest that these off target effects were not affecting our results.

12. A diagram summarizing the proposed pathways downstream of Raptor and their effects would be very helpful towards understanding this complicated system.

A diagram summarizing all the downstream pathways has been added as Supplemental Fig. 15A.

13. The manuscript is disorganized and difficult to understand. Data are presented haphazardly and not in temporal or logical order. The text at times contradicts itself, in some places saying the effect on mass is exclusively through apoptosis and in other places noting a marked reduction in proliferation. Paragraph structure does not lead to easy understanding of the main results. Consider breaking Results section into more subsections to help organize it. The introduction needs a better description of mTor downstream effectors to help the reader interpret the data.

We have modified extensively the manuscript to organize the data in more logical manner. We also made changes to provide a better description of mTORC1 downstream effectors and now include a new diagram. Finally, we fixed some of the inconsistencies to avoid any contradictions and include more subsections to help organize it.

Minor concerns

1. Bar graph in 1A contains extra 'o's in the y axis. Please clarify meaning of "percentage respect raptor"

"Percentage respect raptor" was a typo, It is now modified to raptor/tubulin or phospho/total (Fold Change)". The extra 'o's were also corrected.

2. Please include the age of mice for the immunostaining in panel 1B (Now Supplemental Fig. 1A). The quality of the image on the pdf for review is not high enough to clearly see the result. Consider enlarging the images, improving the clarity or selecting images that are in focus, and including an insert that shows the odd-looking insulin staining in higher magnification and/or in monochrome like the pS6 panel

The experiments were performed in 60 day-old mice. This information is now included in the Figure legend. Quality of the image was also improved, images were enlarged and insulin staining is now displayed.

3. The age of mice used for MIPcreERT deletion cannot be found in the manuscript. Age of BRAKO islets used for insulin secretion assay in Supplemental Figure 4H (Now Fig. 1O) is not given.

The age of *MIP- β raKO^{ff}* mice and controls have been added to the manuscript and Figure Legend. The ages of the mice used for secretion were: *raptor^{ff}* (60 day-old), *MIP- β aKO^{ff}* (60 day-old; TMX injected at 1 month of age) and *β raKO* (30 day-old; the time when we were still able to obtain islets).

4. The Methods section is disorganized and repetitive. For example, the tamoxifen section is included twice, and there is no description of which mice received tamoxifen injections.

Thank you for the comment. Now, the Methods section has been carefully reviewed and organized. Details about TMX administration have been included.

5. Figure font size too small to read even when blown up to max size on my monitor

Figures and font size have been enlarged.

6. If controls are repeated in Figure 2E and F this should be stated in the legend

Controls for CQ and 3-MA are the same. Figure was split for simplicity. This is now included in the Figure Legend.

7. Supplemental Figure 5D (Now Fig. 3I) concentration of glucose used should be included in the legend rather than just 'low' and 'high'

This is now included.

8. Typographical errors: line 87: form, fig 2A 'clev', line 120 'increase',

Fixed.

References:

1. Watada H, Fujitani Y. Minireview: Autophagy in pancreatic beta-cells and its implication in diabetes. *Mol Endocrinol* **29**, 338-348 (2015).
2. Ebato C, *et al.* Autophagy is important in islet homeostasis and compensatory increase of beta cell mass in response to high-fat diet. *Cell Metab* **8**, 325-332 (2008).
3. Kuma A, *et al.* The role of autophagy during the early neonatal starvation period. *Nature* **432**, 1032-1036 (2004).
4. Thoreen CC, Sabatini DM. Rapamycin inhibits mTORC1, but not completely. *Autophagy* **5**, 725-726 (2009).
5. Choo AY, Yoon SO, Kim SG, Roux PP, Blenis J. Rapamycin differentially inhibits S6Ks and 4E-BP1 to mediate cell-type-specific repression of mRNA translation. *Proc Natl Acad Sci U S A* **105**, 17414-17419 (2008).
6. Klionsky DJ, Meijer AJ, Codogno P. Autophagy and p70S6 kinase. *Autophagy* **1**, 59-60; discussion 60-51 (2005).
7. Datan E, *et al.* mTOR/p70S6K signaling distinguishes routine, maintenance-level autophagy from autophagic cell death during influenza A infection. *Virology* **452-453**, 175-190 (2014).
8. Armour SM, Baur JA, Hsieh SN, Land-Bracha A, Thomas SM, Sinclair DA. Inhibition of mammalian S6 kinase by resveratrol suppresses autophagy. *Aging (Albany NY)* **1**, 515-528 (2009).
9. Scott RC, Schuldiner O, Neufeld TP. Role and regulation of starvation-induced autophagy in the *Drosophila* fat body. *Dev Cell* **7**, 167-178 (2004).
10. Mammucari C, *et al.* FoxO3 controls autophagy in skeletal muscle in vivo. *Cell Metab* **6**, 458-471 (2007).
11. Blandino-Rosano M, *et al.* 4E-BP2/SH2B1//IRS2 are part of a novel feedback loop that controls beta-cell mass. *Diabetes*, (2016).
12. Elghazi L, *et al.* Decreased IRS signaling impairs beta-cell cycle progression and survival in transgenic mice overexpressing S6K in beta-cells. *Diabetes* **59**, 2390-2399 (2010).
13. Klionsky DJ, *et al.* Guidelines for the use and interpretation of assays for monitoring autophagy (3rd edition). *Autophagy* **12**, 1-222 (2016).

14. Ling H, *et al.* Role of lysosomal cathepsin activities in cell hypertrophy induced by NH₄Cl in cultured renal proximal tubule cells. *J Am Soc Nephrol* **7**, 73-80 (1996).

REVIEWERS' COMMENTS:

Reviewer #1 (Remarks to the Author):

- New Figure 3C is confusing. Differences among images are hard to see, and some controls are missing (control mice treated with CQ/3MA).
- Please correct the new order of supplemental figures, (supplemental figure 14 is referenced after supplemental 1, supplemental #15 is referenced before #13 and #14...)
- The authors state that caS6K on the background of bRaKo reverses the autophagic phenotype. A single static measurement of LC3 I/II (in Supplemental Fig. 9A) cannot support this statement (DOI: 10.1080/15548627.2015.1100356). The authors show convincing evidences of a reversion in the cell death phenotype, but whether autophagy plays a role in the caS6K phenotype is not clear.
- There are statements in the revised version of the manuscript regarding the role of Raptor during embryonic development such as:
"Disruption of mTORC1 signaling during development failed to impair β -cell expansion as β raKO mice were born with normal β -cell mass (Supplemental Fig. 2A-D)" "The results obtained with the β raKO mice indicate that mTORC1 is critical to maintenance of postnatal β -cell mass but not during embryonic development."

The authors have not explored the role of Raptor during development, as they used RIP-Cre, which is expressed in developed beta cells (recombination occurs perinatally). Perhaps mTORC1 plays a role in setting the beta cell mass during development (for example: by contributing to the expansion of pancreatic progenitors), but the authors did not explore this. In order to support the aforementioned statements, the authors should have used other systems that allow earlier recombination, such as Pdx1-Cre or Ngn3-Cre.

Reviewer #3 (Remarks to the Author):

The authors adequately addressed previous comments. This is an important contribution to our current efforts on understanding the mechanisms of beta cell dysfunction and reduced beta cell mass in diabetes.

Reviewer #4 (Remarks to the Author):

My concerns have been addressed. This is a nice, detailed genetic study into the complicated impact of mTOR signaling pathways on mouse islet mass and function in vivo.

Reviewers' comments:

Reviewer 1

- *New Figure 3C is confusing. Differences among images are hard to see, and some controls are missing (control mice treated with CQ/3MA).*
- We added all the controls images to new Supplementary Fig. 9. We were forced to do this due to the lack of space on Figure 3.
-
- *Please correct the new order of supplemental figures, (supplemental figure 14 is referenced after supplemental 1, supplemental #15 is referenced before #13 and #14...)*
- The order of the Supplementary Figs. has been updated accordingly and also to include the images of the most important gels as requested. Thank you.
-
- *The authors state that caS6K on the background of bRaKo reverses the autophagic phenotype. A single static measurement of LC3 I/II (in Supplemental Fig. 9A) cannot support this statement (DOI: 10.1080/15548627.2015.1100356). The authors show convincing evidences of a reversion in the cell death phenotype, but whether autophagy plays a role in the caS6K phenotype is not clear.*

The statements related to S6K and autophagy in Results and Discussion have been modified to avoid mentioning that autophagy plays a role in the caS6K phenotype. Now they read as follow,

Results

Page 10

“The levels of LC3-II were restored to normal in *braKO;caS6K* and *braKO;caS6K;Eif4ebp2^{-/-}* mice (Supplementary Fig. 12a). Surprisingly, this effect was not mediated by ULK phosphorylation since no changes were observed in islets of *caS6K*”

Discussion

Page 14.

“The genetic reconstitution studies in *braKO* mice demonstrated that downstream of mTORC1 (Supplementary Fig. 20a): 1) ULK1 and S6K axis control β -cell survival, 2) S6K mediates the effects on β -cell size and 3) 4E-BP2/eIF4E (and not 4E-BP1) induces β -cell proliferation.”

Page 15.

“The potential effect of S6K on regulation of β -cell autophagy is interesting and our experiments are consistent with a model in which mTORC1/S6K inhibits autophagy by increasing FLIP_L levels and this process ultimately inhibits autophagy-mediated cell death by binding to ATG3 and decreasing LC3”

- There are statements in the revised version of the manuscript regarding the role of Raptor during embryonic development such as:

*“Disruption of mTORC1 signaling during development failed to impair β -cell expansion as *braKO* mice were born with normal β -cell mass (Supplemental Fig. 2A-D)” “The results obtained with the *braKO* mice indicate that mTORC1 is critical to maintenance of postnatal β -cell mass but not during embryonic development.”*

*The authors have not explored the role of Raptor during development, as they used RIP-Cre, which is expressed in developed beta cells (recombination occurs perinatally). Perhaps mTORC1 plays a role in setting the beta cell mass during development (for example: by contributing to the expansion of pancreatic progenitors), but the authors did not explore this. In order to support the aforementioned statements, the authors should have used other systems that allow earlier recombination, such as *Pdx1-Cre* or *Ngn3-Cre*.*

We agree with the reviewer that those statements were confusing. We have modified these sentences as follows:

Page 5.

*“*braKO* mice showed progressive reduction in β -cell mass by decreases in proliferation, survival and cell size. β -cell mass is a critical determinant for glucose homeostasis in rodents and humans. *braKO* mice were born with normal β -cell mass (Supplementary Fig. 4a-d).”*

Page 6.

*“The results obtained with the *braKO* mice indicate that mTORC1 is critical to maintenance of postnatal β -cell mass.”*